# VisuRiddles: Fine-grained Perception is a Primary Bottleneck for Multimodal Large Language Models in Abstract Visual Reasoning

**Hao Yan**[1], **Xingchen Liu**[1], **Hao Wang**[2], **Zhenbiao Cao**[1], **Handong Zheng**[1], **Liang Yin**[1]
**Xinxing Su**[2], **Zihao Chen**[2], **Jihao Wu**[2], **Minghui Liao**[2,*] **Chao Weng**[2]
**Wei Chen**[1], **Yuliang Liu**[1,*], **Xiang Bai**[1]
[1]Huazhong University of Science and Technology, [2]Huawei Inc.
{haoyan, xcliu04, caozhenbiao, hdzheng, liangyin, lemuria_chen, ylliu, xbai}@hust.edu.cn
{wanghao.ai, xinxingsu, mhliao}@foxmail.com, {chenzihao23, wujihao}@huawei.com, chao.weng@gmail.com

## Abstract

Recent strides in multimodal large language models (MLLMs) have demonstrated significant progress in many reasoning tasks, but they still fail in Abstract Visual Reasoning (AVR) tasks. Our experimental findings indicate that the core bottleneck lies not only in the reasoning capabilities of MLLMs but more critically in their absence of fine-grained perception. To address this issue, we present VisuRiddles, a dedicated resource for AVR research. It consists of (i) a benchmark, collected from real-world data, for the systematic evaluation of MLLMs' AVR capabilities, and (ii) a synthesizer, which automatically generates AVR instances enriched with perceptual descriptions and reasoning chains, enabling supervised training and deeper investigation. Building on VisuRiddles, we propose a two-stage training paradigm that progressively enhances perceptual ability and strengthens reasoning, producing the Perception-Augmented Visual Reasoner (PAVR). Experiments demonstrate that PAVR unifies perception and reasoning, substantially outperforming both open-source and commercial MLLMs, thereby underscoring fine-grained perception as the primary bottleneck in AVR. Code and data are available at https://github.com/yh-hust/VisuRiddles

## 1 Introduction

Reasoning is a core component of human intelligence (Johnson-Laird, 2010). Enhancing this capability in Multimodal Large Language Models (MLLMs) is key to bridging the gap to human-level performance (Schulze Buschoff et al., 2025; Caffagni et al., 2024). Recent advancements (Bai et al., 2025; Chen et al., 2024a; Wu et al., 2024; Yu et al., 2024a;b; Li et al., 2024) in MLLMs have made significant progress in various general visual tasks, but still struggle with Abstract Visual Reasoning (AVR) tasks (Mitchell et al.). As shown in Fig. 1(a), even advanced models such as Gemini2.5 Pro continue to face significant challenges in this task, showing a substantial gap compared to human performance. While such AVR tasks are relatively easy for humans, even the most advanced MLLMs struggle to solve them. It unexpected that models with strong performance in general understanding and reasoning tasks still fall short in handling such reasoning challenges. In fact, the challenges of AVR primarily stem from two aspects: fine-grained perception and reasoning. Compared to the recent advances in enhancing reasoning capabilities, the ability to perceive subtle visual structures, such as position, style, and attribute, has been largely overlooked in existing studies (Xu et al., 2025b; Jiang et al., 2024). Unlike humans, which naturally process such fine-grained visual cues, MLLMs often lack the perceptual ability required for abstract graphics understanding (Tong et al., 2024; Cao et al., 2024). As illustrated in Fig. 1(b), once abstract graphics are reformulated into structured perceptual descriptions, the model is able to generate accurate responses. This highlights that strengthening perceptual capability is a crucial factor for advancing AVR. Therefore, genuine progress in enabling MLLMs to master AVR requires addressing both perception and reasoning, with special emphasis on enhancing the often-overlooked perceptual capability.

---

[*]Corresponding authors.

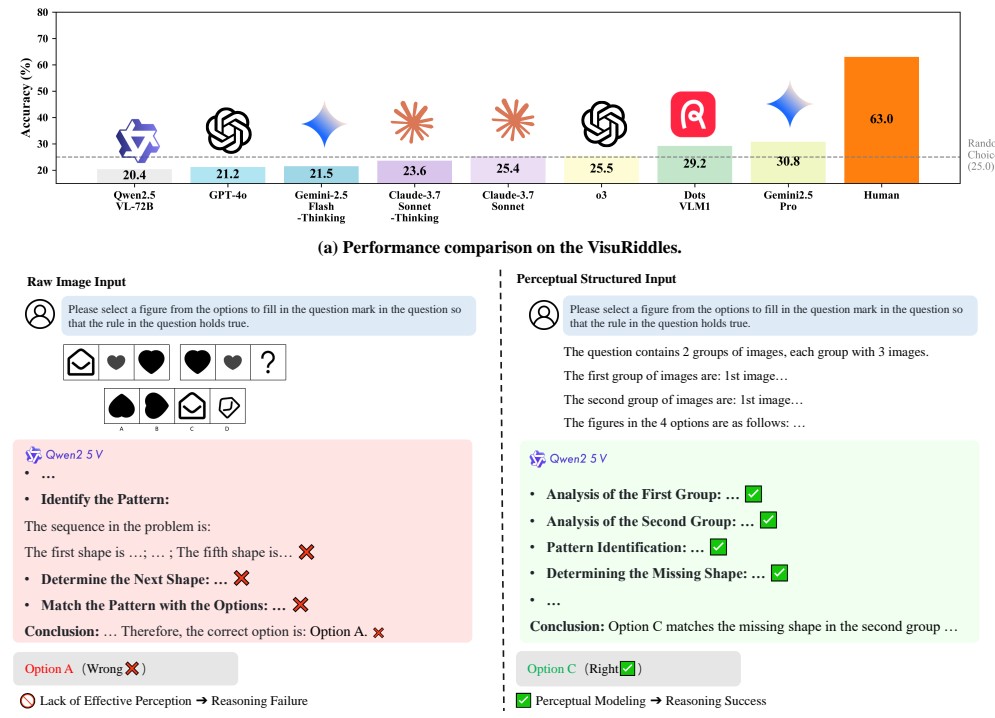

(a) Performance comparison on the VisuRiddles.

(b) Comparison between abstract graphics and perceptual description input.

Figure 1: Performance and analysis of MLLMs on AVR. (a) Most advanced MLLMs achieve limited accuracy on VisuRiddles, often close to random choice and far below human performance. (b) Model responses to abstract graphics and their perceptual descriptions show that once equipped with perceptual capability, MLLMs can succeed in AVR task.

To advance the study of AVR tasks, we introduce the VisuRiddles benchmark, which is primarily derived from real riddles and enables objective evaluation of MLLMs' performance. Since existing datasets lack fine-grained perceptual annotations, we further design the VisuRiddles Synthesizer, which automatically generates AVR instances enriched with structured perceptual descriptions. Leveraging this synthesized data, we perform Supervised Fine-Tuning (SFT) to enhance the model's ability to perceive fine-grained visual cues, thereby equipping it with the perceptual foundation necessary for AVR. Although SFT substantially improves the perceptual capability of MLLMs, they still face persistent bottlenecks in AVR, including errors in perceptual strategy (e.g., difficulty in determining whether the appropriate interpretive cue lies in the figure's symmetry axis or in its right-angle structures) and limited reasoning ability on more difficult instances. Therefore, we introduce Reinforcement Learning (RL), which leverages the enhanced perceptual capability to guide models toward more reliable perceptual grounding and further strengthen their reasoning ability.

Building upon these components, we develop the Perception-Augmented Visual Reasoner (PAVR), which unifies enhanced perceptual capability with improved reasoning ability to address the dual challenges of AVR. Experimental results show that PAVR significantly outperforms advanced commercial models on AVR tasks, highlighting a systematic solution to AVR tasks.

Our contributions can be summarized as follows:

(1) We introduce VisuRiddles, which comprises a benchmark derived from real riddles for objectively evaluating MLLMs on AVR tasks, and a synthesizer that automatically generates AVR instances enriched with structured perceptual descriptions.

(2) We leverage synthesized data for SFT to enhance fine-grained perceptual capability for AVR tasks, and further adopt RL to improve reasoning ability and stabilize perceptual strategy selection, resulting in the Perception-Augmented Visual Reasoner.

(3) We conduct extensive experiments to demonstrate our key findings, including the reasoning limitations of current MLLMs, the effectiveness of our synthesized riddles, and the effective approach to addressing abstract visual problems.

## 2 RELATED WORK

### 2.1 MULTIMODAL BENCHMARKS

High-quality benchmarks are essential for driving progress in MLLMs. Earlier evaluation effort predominantly centered on basic visual comprehension tasks, leading to the development of extraction-oriented benchmarks (Mathew et al., 2021; Masry et al., 2022; Liu et al., 2024a). These benchmarks primarily assess a model's ability to identify and align explicit visual elements. However, they depend heavily on surface-level cues, offering little assessment of an MLLM's capability for abstraction and logical reasoning. Therefore, general-purpose benchmarks (Xu et al., 2023; Bitton et al., 2023; Yang et al., 2024; Yue et al., 2024; 2025) are widely focused on. These benchmarks span multiple tasks, such as visual understanding and mathematical reasoning, and are designed to assess the generalist capabilities of MLLMs. However, they tend to emphasize knowledge-based evaluation over visual logical reasoning. Moreover, due to their strong semantic dominance, MLLMs may directly rely on their inherent knowledge, making it difficult to disentangle and evaluate models' true visual reasoning capabilities. Therefore, some studies have shifted toward evaluating visual logical reasoning, leading to logic-oriented benchmarks such as RAVEN (Zhang et al., 2019), I-RAVEN (Hu et al., 2021), RAVEN-FAIR (Benny et al., 2021), CVR (Zerroug et al., 2022), RPMs (Zhang et al., 2024b), MaRs-VQA (Cao et al., 2024), MathVerse (Zhang et al., 2024a), MARVEL (Jiang et al., 2024), PuzzleVQA (Chia et al., 2024), VisCogBench (Cao et al., 2024), VisuLogic (Xu et al., 2025b), VisualPuzzle (Song et al., 2025), VisualSphinx (Feng et al., 2025) and MV-MATH (Wang et al., 2025a). These benchmarks use structured diagrams, mathematical forms, and spatial layouts to assess analogical reasoning, compositionality, and pattern induction, signaling a move toward higher-level AVR. However, these benchmarks still exhibit notable limitations, particularly in their partial reliance on external knowledge and lack breadth in reasoning coverage.

### 2.2 MULTIMODAL LARGE LANGUAGE MODELS FOR VISUAL REASONING

Advanced commercial LLMs, including the GPT (OpenAI, 2022), Gemini (Team et al., 2023), and Claude (Anthropic, 2024) series, have exhibited impressive capabilities in multimodal understanding and generation (Yin et al., 2024; Yan et al., 2023). These advances have spurred systematic research on MLLMs in both academia and industry. Early representative general-purpose MLLMs, such as LLaVA (Liu et al., 2023), Instruct-BLIP (Dai et al., 2023), Qwen-VL (Bai et al., 2023), and Intern-VL (Chen et al., 2024b), perform well on general multimodal tasks but exhibit limitations in fine-grained visual perception. Subsequently, DeepSeek-VL (Lu et al., 2024), Monkey (Li et al., 2024), TextMonkey (Liu et al., 2024b), InternLM-XComposer2-4KHD (Dong et al., 2024), Qwen2-VL (Wang et al., 2024a) have introduced high-resolution visual perception mechanisms, establishing a more robust foundation for fine-grained visual understanding and reasoning. However, such improvements primarily expand the perceptual scope of MLLMs without substantially enhancing their reasoning capabilities. Therefore, inference-time scaling methods including MMVP (Tong et al., 2024), LLaVA-CoT (Xu et al., 2024), BBA (Zhao et al., 2024), R-CoT (Deng et al., 2024), and RedStar (Xu et al., 2025a) have introduced and advanced inference-time scaling techniques such as chain-of-thought (CoT), significantly enhancing the visual reasoning capabilities of MLLMs. More recently, following the advances of Kimi K1.5 (Team et al., 2025) and DeepSeek-R1 (Guo et al., 2025), a series of studies, such as R1-OneVision (Yang et al., 2025), LMM-R1 (Peng et al., 2025), MM-EUREKA (Meng et al., 2025), R1-V (Chen et al., 2025), Visual-RFT (Liu et al., 2025b), VisualPRM (Wang et al., 2025b), OThink-MR1 (Liu et al., 2025a), and VLM-R1 (Shen et al., 2025), have employed reinforcement learning to develop adaptive reasoning policies and improve generalization in complex visual reasoning tasks.

Table 1: The statistics of VisuRiddles, including the number questions, average question length (in tokens), number of associated images, and the corresponding answer format.

| Type | Questions Number | Question length | Answer Formats |
|---|---|---|---|
| Sudoku | 100 | 48.3 | Constraint-based grid |
| RAVEN | 100 | 70.7 | Single-choice |
| Numerical | 250 | 26.3 | Single-choice |
| Stylistic | 117 | 26.3 | Single-choice |
| Attribute | 97 | 27.0 | Single-choice |
| Positional | 111 | 25.5 | Single-choice |
| Spatial | 156 | 27.2 | Single-choice |
| Other | 69 | 30.3 | Single-choice |
| Total | 1000 | 32.5 | - |

## 3 VISURIDDLES

In this section, we introduce **VisuRiddles**, a innovative and comprehensive resource for AVR research. It consists of the **VisuRiddles Benchmark** (Sec. 3.1), built from real-world problems to objectively evaluate MLLMs' AVR performance, and the **VisuRiddles Synthesizer** (Sec. 3.2), designed to generate diverse perceptually annotated AVR instances for model training.

### 3.1 VISURIDDLES BENCHMARK

Unlike RAVEN (Zhang et al., 2019) and PuzzleVQA (Chia et al., 2024), which focus on general visual reasoning tasks, we target the most challenging part of AVR tasks. Thus, the VisuRiddles benchmark has been constructed to evaluate MLLMs. It explicitly evaluates models across five key dimensions: a) **Numerosity** assesses the model's ability to perceive and reason about quantity and distribution; b) **Attribute** evaluates the understanding of intrinsic visual features that determine structural semantics; c) **Style** tests the capability to identify and generalize transformation-based visual patterns; d) **Position** reflects the ability to reason over the relative positions and layout of visual elements; e) **Spatiality** examines the understanding of three-dimensional structures, shape variations, and spatial transformations in abstract graphics. To extend the evaluation beyond low-level perceptual reasoning and address limitations in task diversity, structural complexity, and the inherent uncertainty introduced by single-choice formats, we additionally incorporate two high-level reasoning tasks that require analogical abstract reasoning and consistency-based logical reasoning, as represented by **RAVEN** and **Sudoku**, respectively. Finally, VisuRiddles also includes a subset, namely Other of AVR tasks involving planar shape composition and character-based semantic patterns. VisuRiddles provides a unified benchmark spanning basic to high-level reasoning, enabling comprehensive assessment of AVR, with representative samples shown in Fig. 5. We construct the VisuRiddles benchmark through a structured pipeline comprising **Collection**, **Cleaning**, and **Consolidation**. We provide the representative examples and construction details in Appendix A.2.

We initially collected 1,275 AVR problems, each accompanied by expert-level analysis and answer. After careful filtering and quality control, we construct VisuRiddles with 800 samples in the basic categories and 200 samples in the high-level categories. Key statistics of VisuRiddles are summarized in Tab. 1. For the basic AVR tasks, each question is formulated as a single-choice problem, with answer option distributions of A (25.75%), B (24.75%), C (24.375%), and D (25.125%). In the high-level categories, RAVEN Reasoning and Sudoku Reasoning each account for 10% of the total data, and require models to generate exact symbolic outputs to be considered correct.

### 3.2 VISURIDDLES SYNTHESIZER

Recent MLLMs demonstrate strong general reasoning, but their limited performance on AVR is largely attributed to insufficient capability of fine-grained perception for abstract graphics. Most existing datasets lack intermediate perceptual description, providing only question-answer pairs (Xu et al., 2025b; Jiang et al., 2024; Song et al., 2025). This prevents explicit modeling of the perception-to-reasoning process, leading to black-box inference, weak inductive capability, and poor generalization.

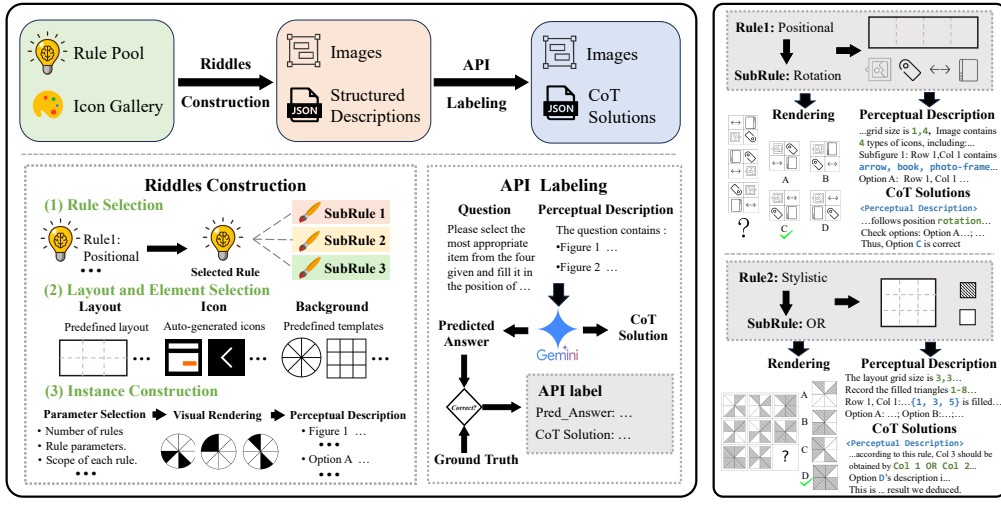

(a) Unified Pipeline for Riddles Synthesis    (b) Instantiated Examples

Figure 2: Overview of the **VisuRiddles Synthesizer**. (a) A unified pipeline for generating abstract graphics with fine-grained perceptual descriptions. (b) Visualization of synthesized riddles based on positional rule and stylistic rule.

More critically, existing commercial models struggle to annotate perceptual processes effectively, and manual labeling requires significant expert effort, both of which further exacerbate the issue. To address this, we introduce **VisuRiddles Synthesizer**, a Riddles Synthesis Framework to provides abstract graphics with aligned fine-grained perceptual description.

Figure 2 illustrates the pipeline of our VisuRiddles Synthesizer for AVR task. The VisuRiddles Synthesizer pipeline comprises two stages: Riddles Construction and API Labeling. The former generates abstract visual instances with aligned perceptual descriptions, while the latter generates reasoning chains based on these descriptions. Details and representative examples are provided in Appendix B.2 and Appendix B.3.

The VisuRiddles Synthesizer generates data that spans five core perceptual reasoning types and two high-level reasoning tasks. The detailed configurations of VisuRiddles Synthesizer are provided in Table 5. The VisuRiddles Synthesizer not only provides riddles' visual instances but also structured perceptual descriptions, thereby enhancing the model's fine-grained perceptual ability for AVR tasks. It is noteworthy that the riddles instances generated by the VisuRiddles Synthesizer focus on improving fine-grained perception, rather than reasoning. Therefore, their reasoning difficulty is deliberately kept lower than that of real-world riddles.

## 4    PERCEPTION-AUGMENTED VISUAL REASONER

To tackle the challenges of AVR, we introduce the Perception-Augmented Visual Reasoner (PAVR). As illustrated in Fig. 3, PAVR is developed through a two-stage training paradigm. In the first stage, SFT equips the model with the ability to capture fine-grained visual cues in abstract graphics. Building upon this, the second stage employs RL to effectively integrate these visual cues into stable reasoning strategies for solving complex tasks, resulting in PAVR. These two stages are complementary: SFT enables the model to see clearly, while RL guides it to reason reliably.

**Perceptual Augment via Supervised Fine-Tuning.** The lack of fine-grained perceptual ability remains a major bottleneck for MLLMs on AVR tasks, yet existing real-world datasets typically provide only single-choice answer and rarely provide explicit perceptual annotations to supply sufficient supervision. To address this limitation, we employ the *VisuRiddles Synthesizer* described in Sec. 3.2 to generate AVR instances across seven categories, each enriched with structured perceptual

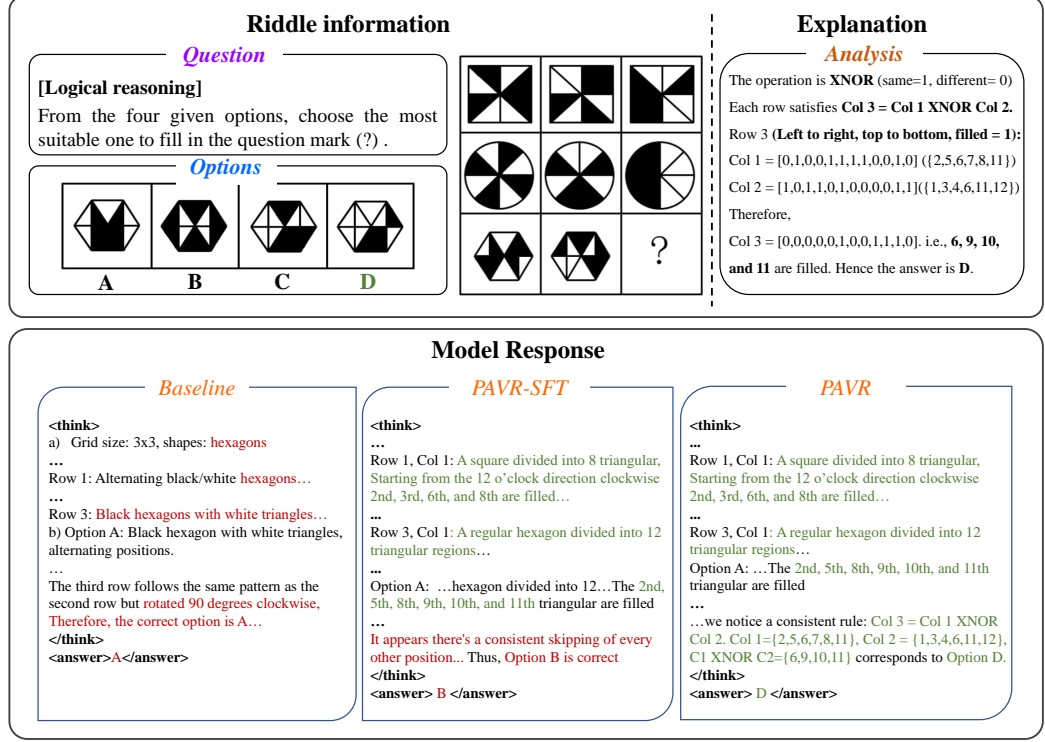

Figure 3: Overview of PAVR. (i) Baseline: Incorrect perception leads to incorrect results. (ii) PAVR-SFT, which is trained on synthesized data to enhance fine-grained perceptual ability, can accurately understand visual content in riddles but fails in pattern identification. (iii) PAVR, which builds upon PAVR-SFT with reinforcement learning, can effectively recognize patterns and derive the correct answer.

descriptions. These synthesized examples serve as training data for SFT, enhancing the model's perceptual ability and enabling it to capture fine-grained cues in abstract graphics. This establishes a reliable foundation for subsequent reasoning optimization through reinforcement learning.

**Reasoning Optimization via Reinforcement Learning.** Although SFT on synthesized data improves fine-grained perceptual ability, it still faces limitations in AVR, including unstable perceptual strategy selection (e.g., axis vs. angles) and insufficient reasoning on complex tasks. Therefore, we apply Group Relative Policy Optimization (GRPO) (Guo et al., 2025) to endow the model with more reliable perceptual grounding and stronger reasoning ability. The reward design includes: (i) Answer Reward, giving 1 for correct answers and 0 otherwise, and (ii) Format Reward, incentivizing outputs that match the required template: <think> ... </think> <answer> ... </answer>. The training data for GRPO is likewise generated by the *VisuRiddles Synthesizer*. This process enables the model to achieve more reliable perceptual grounding and stronger reasoning capabilities.

## 5 EXPERIMENT

### 5.1 EXPERIMENTAL SETUP

We use Qwen2.5-VL-7B (Bai et al., 2025) as the baseline model to develop PAVR. In the supervised fine-tuning stage, the model is trained for 20 epochs using 20K synthesized AVR instances generated by the VisuRiddles Synthesizer. For this stage, we employed the AdamW Optimizer, with a Batch Size of 16, and a Learning Rate set at $5e-7$. Subsequently, in the GRPO stage, optimization is performed using 4K synthesized instances over 40 epochs. For this stage, the Learning Rate was set to $1e-6$, the Rollout Number was 5, and the KL Loss Coefficient and CLIP Ratio were 0.01 and

Table 2: Evaluation result on VisuRiddles benchmark. The superscript number indicates the number of answer options (e.g., "4" means a 4-choice question with one correct option; * represents a vast combinatorial solution space). The best results are marked **bold** and the second results are underlined.

| Model | Param | Num[4] | Styl[4] | Attr[4] | Posit[4] | Spat[4] | Sudo[*] | Rav[8] | Other[4] | Avg |
|---|---|---|---|---|---|---|---|---|---|---|
| Human | - | 61.3 | 60.9 | 67.5 | 67.9 | 58.8 | - | - | 61.9 | - |
| *Open-Source MLLMs* | | | | | | | | | | |
| Minicpm-V-2.6 | 8B | 21.6 | 26.5 | 23.7 | 31.5 | 23.1 | 0.0 | 10.0 | 24.6 | 20.6 |
| InternVL2.5-8B | 8B | 21.2 | 13.7 | 30.9 | 23.4 | 23.1 | 0.0 | 0.0 | 29.0 | 18.1 |
| InternVL2.5-8B-MPO | 8B | 22.0 | 23.9 | 27.8 | 25.2 | 21.8 | 0.0 | 3.0 | 27.5 | 19.4 |
| Deepseekvl2 | 27B | 21.2 | 27.4 | 18.6 | 18.0 | 17.3 | 0.0 | 15.0 | 13.0 | 17.4 |
| InternVL2.5-38B | 38B | 22.0 | 19.7 | 29.9 | 26.1 | 29.5 | 1.0 | 19.0 | 30.4 | 22.3 |
| InternVL2.5-38B-MPO | 38B | 26.0 | 20.5 | 19.6 | 27.0 | 27.6 | 0.0 | 18.0 | 31.9 | 22.1 |
| Qwen2.5VL-32B | 32B | 23.2 | 20.5 | 30.9 | 27.0 | 29.5 | 0.0 | 40.0 | 24.6 | 24.5 |
| InternVL2.5-78B | 78B | 26.0 | 25.6 | 27.8 | 29.7 | 23.7 | 0.0 | 16.0 | 27.5 | 22.7 |
| InternVL2.5-78B-MPO | 78B | 25.6 | 30.8 | 24.7 | 27.9 | 26.3 | 0.0 | 12.0 | 20.3 | 22.2 |
| InternVL2.5-78B-MPO(cot) | 78B | 23.2 | 23.1 | 24.7 | 28.8 | 22.4 | 0.0 | 11.0 | 23.2 | 20.3 |
| Qwen2.5VL-72B | 72B | 23.6 | 23.1 | 19.6 | 30.2 | 26.9 | 0.0 | 62.0 | 23.9 | 25.9 |
| Qwen2.5VL-72B(cot) | 72B | 27.2 | 26.5 | 24.7 | 19.8 | 28.2 | 0.0 | 55.0 | 23.2 | 26.0 |
| Qwen3-VL-235B-Instruct | 235B | 31.2 | 28.2 | 36.1 | 34.2 | 26.3 | 29.0 | 53.0 | 27.5 | 32.6 |
| Qwen3-VL-235B-Thinking | 235B | 31.2 | 29.9 | 44.3 | 33.3 | 30.1 | 33.0 | 49.0 | 39.1 | 34.9 |
| Dots.vlm1 | 671B | 30.4 | 31.6 | 38.1 | 38.7 | 35.3 | 1.0 | 20.0 | 33.3 | 29.2 |
| *Commercial products* | | | | | | | | | | |
| GPT-4o | - | 23.2 | 21.4 | 29.9 | 27.9 | 28.2 | 0.0 | 14.0 | 24.6 | 21.8 |
| GPT-4o(cot) | - | 22.4 | 23.9 | 32.0 | 29.7 | 30.1 | 0.0 | 22.0 | 29.0 | 23.7 |
| o3 | - | 28.4 | **40.2** | 41.2 | 27.0 | 21.8 | 0.0 | 25.0 | 33.3 | 27.0 |
| claude-3-7-sonnet | - | 24.0 | 27.4 | 27.8 | 29.7 | 28.2 | 4.0 | 43.0 | 31.9 | 26.5 |
| claude-3-7-sonne(cot) | - | 24.4 | 36.8 | 27.8 | 29.7 | 25.6 | 3.0 | 36.0 | 27.5 | 26.2 |
| claude-3-7-sonnet-thinking | - | 24.0 | 24.8 | 33.0 | 23.4 | 25.6 | 3.0 | 24.0 | 26.1 | 23.2 |
| Gemini-2.5-flash-thinking | - | 21.2 | 30.8 | 27.8 | 21.6 | 16.7 | 17.0 | 16.0 | 23.2 | 21.5 |
| Gemini2.5-pro | - | 31.6 | 31.6 | 48.5 | 26.1 | 30.1 | 39.0 | 30.0 | 44.9 | 33.9 |
| GPT-5 | - | 30.8 | 30.8 | 38.1 | 32.4 | 30.8 | 2.0 | 29.0 | 31.9 | 28.7 |
| *Ours* | | | | | | | | | | |
| Baseline(Qwen2.5VL-7B) | 7B | 24.4 | 28.2 | 23.7 | 22.5 | 25.0 | 0.0 | 48.0 | 24.6 | 24.6 |
| PAVR-SFT | 7B | 31.2 | 31.6 | 44.3 | 31.5 | 45.5 | 43.0 | 61.0 | 39.1 | 39.5 |
| PAVR | 7B | **39.6** | 39.3 | **50.5** | **39.6** | **51.9** | **46.0** | **65.0** | **55.1** | **46.8** |

1.0, respectively. All experiments are conducted on a setup comprising 8 NVIDIA A800 80G GPUs. Evaluation settings are summarized in Appendix C.

## 5.2 MAIN RESULTS

Table 2 presents performance of various MLLMs on VisuRiddles benchmark. We summarize the key observations as follows:

**Performance of Open-Source MLLMs.** Most open-source MLLMs demonstrate limited performance on VisuRiddles, with accuracies across core categories often close to random choice and consistently failing on the two high-level reasoning tasks, which reveals a substantial gap between current model capabilities and human performance in AVR.

**Performance of Commercial Models with Thinking Mode.** Several commercial MLLMs with 'thinking' mode achieve comparatively improved performance on VisuRiddles. Their advantage is particularly notable in structured tasks like Sudoku and Raven. These results indicate that AVR can benefit from advanced reasoning capabilities. However, the overall performance remains limited.

**Effectiveness of Model Scaling and CoT Prompting.** Strategies such as increasing model size and applying CoT prompting, though commonly used to enhance reasoning, are ineffective for AVR tasks.

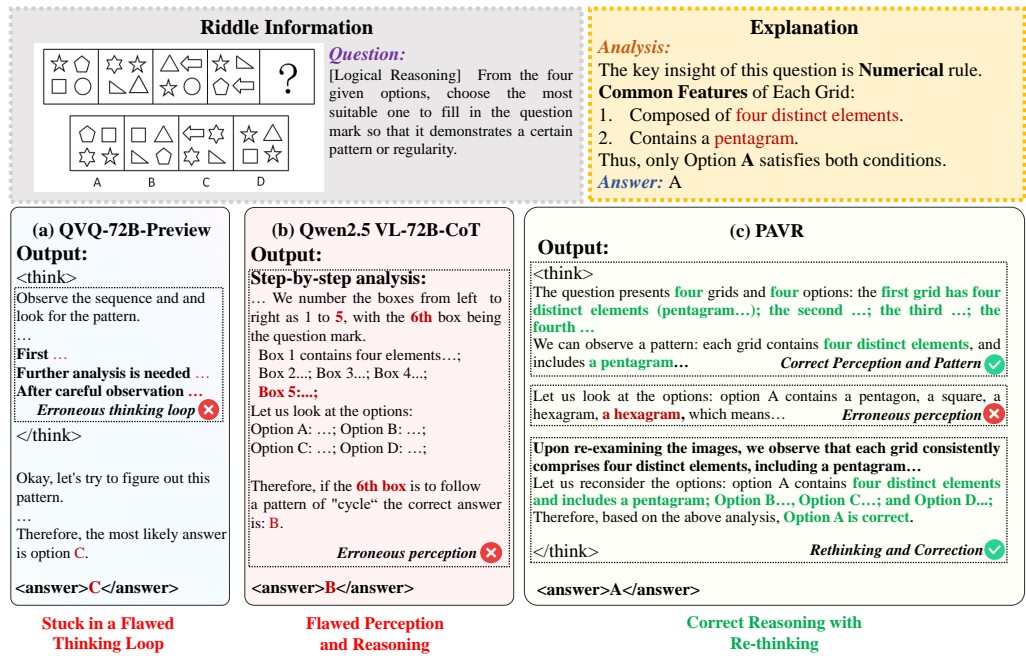

Figure 4: Case study comparing different reasoning strategies on a VisuRiddles example. (a) QVQ-72B reflects a flawed loop under the "thinking" mode. (b) Qwen2.5-VL-72B with CoT prompting exhibits incorrect perceptual understanding. (c) PAVR exhibits accurate perception and coherent reasoning, ultimately arriving at the correct answer.

Models with larger parameter scales do not consistently outperform smaller ones, and CoT yields only limited improvements on select tasks. These findings indicate that scaling in model parameter or inference time is insufficient to address the core challenges of AVR.

**Effectiveness of PAVR.** PAVR significantly outperforms other MLLMs across all categories of VisuRiddles, especially in high-level reasoning tasks, demonstrating the model's ability to handle complex visual patterns. Notably, compared to the baseline, the significant improvement achieved by training on data with perceptual descriptions through SFT highlights the critical role of fine-grained perceptual ability in AVR tasks. On the other hand, the application of RL further enhances performance across various dimensions on VisuRiddles, with the improvements primarily stemming from enhanced reasoning capabilities through strategic guidance.

We also provide the results of PAVR on the VisuLogic (Xu et al., 2025b) benchmark in D. The experimental results further support our conclusions.

## 5.3 VISUALIZATION RESULTS

To further analyze the reason that MLLMs underperform on AVR task, we examine two widely adopted reasoning enhancement strategies: inference-time scaling and CoT prompting. We conduct experiments on QVQ-72B-Preview (equipped with a "think" mode), Qwen2.5-VL-72B (with CoT prompting), and PAVR. Fig. 4 visualizes a representative case of a broadly observed phenomenon uncovered during our experiments. The results reveal clear differences in the reasoning behaviors across these models. The inference-time scaling approach, which leverages "think"-style prompting without any perceptual grounding, frequently leads to verbose yet logically inconsistent reasoning, often ending in incorrect answers. The CoT prompting strategy often suffers from perceptual errors, which cascade into flawed reasoning and ultimately lead to incorrect answers. In contrast, by grounding reasoning in fine-grained perceptual capability, PAVR correctly interprets the riddle and identifies the underlying pattern, ultimately producing the correct answer. Notably, while PAVR occasionally exhibits erroneous perception, we observe a 'rethink' phenomenon, where the model re-examines and corrects them, realigning the reasoning trajectory to reach the correct solution.

These results underscore the importance of perception in AVR. Inference-time scaling and CoT prompting alone cannot compensate for the lack of perceptual grounding, which often leads to incorrect or inconsistent reasoning. Conversely, the incorporation of fine-grained perceptual capability enables the model to capture subtle visual cues and sustain reliable performance on AVR tasks.

## 5.4 ABLATION STUDY

**Ablation Study on Bottlenecks in AVR**. To investigate whether the performance gap in AVR stems from perceptual limitations rather than reasoning ability, we conduct an ablation study using our perception-annotated synthetic data. We evaluate two representative MLLMs, GPT-4o (closed-source) and Qwen2.5VL-72B (open-source), on a subset of our synthetic dataset under two input settings: (i) raw abstract graphics (V), and (ii) structured perceptual descriptions (P).

The experimental results in Tab. 3 show that both models perform poorly when given only raw visual inputs. Taking Sudoku reasoning as an example, this task requires recognizing digit values and their positions within a grid, and inferring a globally consistent solution. Under the Visual setting, both GPT-4o and Qwen2.5VL-72B fail to handle such constraint-based reasoning, while in the perceptual descriptions setting, accuracy increases from near-zero to 15% and 65%, respectively. This improvement primarily stems from structured perceptual descriptions compensating for the MLLMs' limited perceptual abstraction capabilities. The difficulty stems from the nature of Sudoku images: small digits, dense layout, and absent semantic cues make them hard for MLLMs to interpret, though humans process such structure effortlessly through rapid visual scanning.

Table 3: Impact of Descriptions vs. Visual Inputs in MLLMs. Where "V" denotes inputting abstract graphics directly, while "P" refers to replacing graphics with structured perceptual descriptions.

| Model | Num. | Styl. | Attr. | Posit. | Spat. | Sudo. | Rav. | Avg |
|---|---|---|---|---|---|---|---|---|
| GPT-4o(V) | 35.0 | 32.0 | 38.0 | 36.0 | 32.0 | 0.0 | 20.0 | 27.6 |
| GPT-4o(P) | 62.0 (+27.0) | 53.0 (+21.0) | 80.0 (+42.0) | 68.0 (+32.0) | 100.0 (+68.0) | 15.0 (+15.0) | 25.0 (+5.0) | 60.1 (+32.5) |
| Qwen2.5VL(V) | 41.0 | 43.0 | 50.0 | 32.0 | 40.0 | 0.0 | 10.0 | 30.9 |
| Qwen2.5VL(P) | 73.0 (+32.0) | 83.0 (+40.0) | 80.0 (+30.0) | 79.0 (+47.0) | 100.0 (60.0) | 65.0 (+65.0) | 35.0 (+25.0) | 73.6 (+42.7) |

**Ablation Study on Reasoning Enhancements in AVR**. To examine the contribution of reasoning-oriented enhancements, we conduct an ablation study on three components: Caption, CoT, and GRPO, as shown in Table 4. We can draw the following conclusions form the experiments: (i) perception serves as the essential foundation, as reasoning enhancements alone contribute only marginal improvements; (ii) relying solely on structured perceptual descriptions for SFT yields limited gains and weak generalization, whereas augmenting SFT with CoT annotations generated by MLLMs alleviates this issue, endowing the model with fine-grained perceptual ability and partial reasoning capability; and (iii) Building on a perception-grounded model, the incorporation of GRPO further unifies perception and reasoning, ultimately delivering substantial performance improvements.

Table 4: Ablation study on Reasoning Enhancements. Where Caption denotes training with fine-grained perceptual descriptions, CoT denotes augmenting training with chain-of-thought reasoning traces that incorporate structured perceptual descriptions generated by MLLMs, and GRPO refers to applying Group Relative Policy Optimization to strength model's reasoning capability.

| Model | SFT | RL | Avg |
|---|---|---|---|
| Baseline (Qwen2.5-VL) | ✗ | ✗ | 24.6 |
| Baseline + Caption | ✓ | ✗ | 33.3 (+8.7) |
| Baseline + GRPO | ✗ | ✓ | 29.4 (+4.8) |
| Baseline + CoT (PAVR-SFT) | ✓ | ✗ | 39.5 (+14.9) |
| Baseline + CoT + GRPO (PAVR) | ✓ | ✓ | 46.8 (+22.2) |

## 6 CONCLUSION

In this work, we observe that the primary bottleneck of MLLMs in AVR lies in their overlooked fine-grained perceptual capability. To address this issue, we introduce VisuRiddles, a unified resource

that comprises a benchmark for objectively evaluating MLLMs on AVR tasks and a synthesizer for generating training instances with structured perceptual descriptions. Building on this resource, we propose PAVR, which unifies perception and reasoning through supervised fine-tuning for perceptual enhancement and GRPO-based optimization for reasoning improvement for AVR tasks. Experimental results demonstrate that current MLLMs perform poorly on AVR tasks. Moreover, scaling model parameters, CoT prompting, and inference-time scaling fail to effectively address this challenge. These findings highlight that fine-grained perception constitutes the fundamental bottleneck in AVR, while reasoning enhancements applied based on a perception-grounded model can further improve performance.

In short, our work not only offers a systematic resource for both evaluation and training, but also provides a practical framework for addressing the AVR challenge.

**Limitation:** Due to limitations in time and human labor, the scale of the resources (e.g., rule design, icon library) used for data synthesis is restricted, which somewhat limits the diversity and richness of the synthetized data. In future work, we plan to expand the resource pool to a larger scale with greater complexity, further enriching the synthesized riddles.

### ACKNOWLEDGMENT

This work was done during the research internship of Hao Yan, Zhenbiao Cao and Liang Yin at Huawei.

This work was supported by the National Natural Science Foundation of China (NSFC) under Grants No.62576147 and No.62225603.

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

# A DETAILS OF VISURIDDLES BENCHMARK

## A.1 REPRESENTATIVE EXAMPLES OF VISURIDDLES BENCHMARK

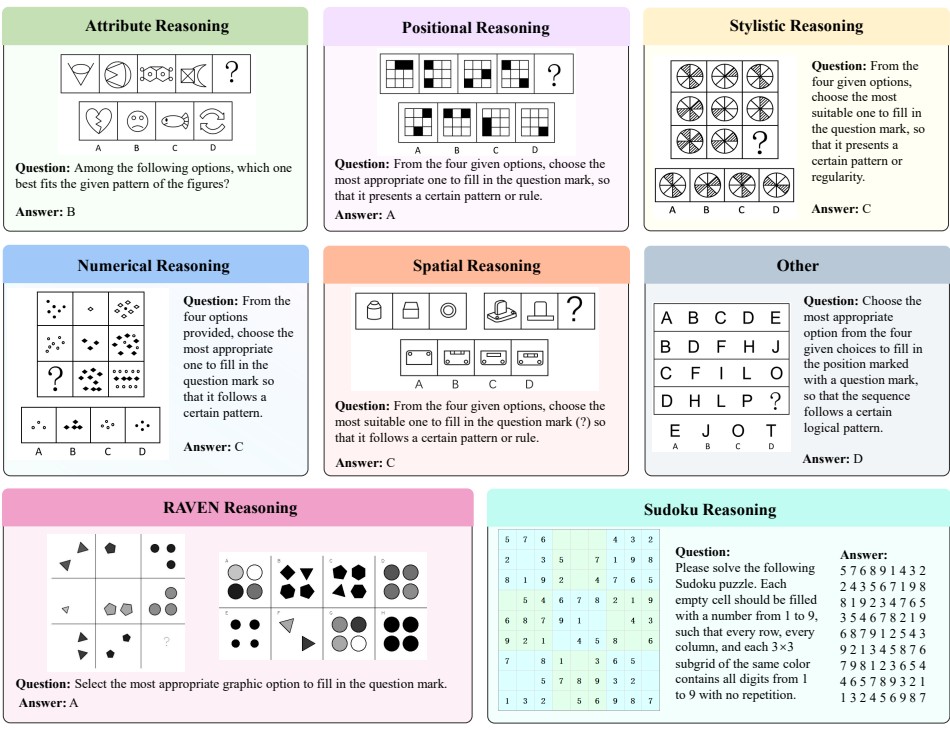

Figure 5: Representative examples from VisuRiddles. The benchmark includes eight reasoning categories, designed to comprehensively evaluate diverse reasoning capabilities of MLLMs.

## A.2 CONSTRUCTION DETAILS OF VISURIDDLES BENCHMARK

We construct the VisuRiddles benchmark through a structured pipeline comprising **Collection**, **Cleaning**, and **Consolidation**:

**Collection.** We manually collect a diverse set of visual reasoning problems from Chinese National Civil Service Examination, which form the five core categories of the benchmark. RAVEN and Sudoku samples are generated following the method[1] and the protocols[2].

**Cleaning.** After initial collection, we recruited twelve trained annotators to verify and refine the dataset. This process involved identifying and removing duplicated questions, correcting answer key errors, and filtering out incomplete or noisy samples to ensure data consistency and reliability. Quality control included annotator training, multi-pass review, dispute resolution, and cross-validation on overlapping samples.

**Consolidation.** All cleaned samples were manually translated into English, covering question and answer. Each item was then categorized into its corresponding reasoning type based on expert explanations provided by the Fenbi educational platform. All data were standardized into a unified JSON format, supporting structured access to image content, metadata, and reasoning annotations.

---

[1]https://github.com/WellyZhang/RAVEN
[2]https://gitee.com/mxx11/sudoku

# B DETAILS OF VISURIDDLES SYNTHESIZER

## B.1 CONFIGURATION OF THE VISURIDDLES SYNTHESIZER

Table 5: Configuration details of the VisuRiddles Synthesizer across seven reasoning categories.

| Rule | Sub-rule | Layout | Template | Icon |
|------|----------|--------|----------|------|
| Numerical | Line, Curve, Angle, Cart, Space, Parts | 1 | 1 | 5k |
| Stylistic | AND, OR, XOR, XNOR | 3 | 5 | 5k |
| Attribute | Element, Group | 2 | 1 | 5k |
| Positional | Translate, Rotate, Flip | 4 | 7 | 5k |
| Spatial | Unfolding, Three-View, 3D-Reconstruction View-Consistency, Multiple-Views | 7 | 13 | 5k |
| Sudoku | Sudoku | 1 | 5 | 9 |
| RAVEN | Raven | 1 | 7 | 10k |

## B.2 SYNTHESIS PIPELINE OF RIDDLES INSTANCES

The synthesis pipeline for generating riddle instances with structured perceptual descriptions consists of two main stages: Instance Construction and API Labeling.

**Instance Construction**
Instance synthesis is responsible for generating the visual instances and corresponding perceptual descriptions. The detailed steps are as follows:

1. Rule Selection:
The synthesis process begins with rule selection. We first extract representative reasoning patterns from real-world standardized tests, such as civil service visual logic exams. These high-level rules are categorized into positional, stylistic, numerical,attribute,spatial,raven and sudoku types, and further decomposed into finer-grained sub-rules, each representing a distinct visual transformation or logical operation (e.g., "simultaneous rotation and revolution", or "black-white XOR operation"). This decomposition enables modular and interpretable synthesis, allowing each instance to target a specific reasoning skill.

2. Element Configuration:
Once a sub-rule is selected, we proceed to define the visual components in a structured manner. This includes: (i) choosing a layout template (e.g., $3 \times 3$ matrix, progression row, etc.), (ii) selecting or auto-generating icon elements with specific visual properties (e.g., shape, color, direction), and (iii) assigning a background grid. Elements are configured in a rule-consistent way: for example, a "positional" sub-rule may require translation across rows, while a "stylistic" sub-rule may apply logical operations to shape fill. Each configuration ensures the pattern remains visually interpretable and solvable.

3. Rendering and Annotating:
Each question instance is constructed by applying parameterized transformations under defined constraints: a single correct answer, consistent application of one rule, and plausible distractors. The resulting sequence of visual instances is rendered into final image grids. In parallel, the framework generates structured **perceptual description** in JSON format, including layout metadata, element-level properties (e.g., object shape, position, state), and reasoning traces. These annotations bridge perception and reasoning, enabling models to explicitly learn visual abstraction rather than relying on black-box pattern matching.

**API Labeling**
Due to the poor performance of existing commercial models in abstract visual reasoning (AVR) tasks, especially when processing images, these models often suffer from significant hallucination issues and fail to generate correct reasoning annotations. Thus, directly feeding images into the model does not produce accurate results. To overcome this, we introduce a new step: first, we replace the images with corresponding perceptual descriptions and input these descriptions into a large model. This allows the model to generate CoT based on the perceptual descriptions. The detailed steps are as follows:

1. CoT Generation:
Perceptual descriptions are first extracted from the synthesized visual instances to supply fine-grained visual information. These descriptions are then provided to an advanced commercial model (Gemini 2.5-Flash-Think[3] in this work), which generates a CoT reasoning process and produces a corresponding answer.

2. Validation:
The generated answer is compared to the correct answer of the synthesized riddle. If the model's output does not align with the correct answer, the data point is discarded, ensuring that only valid and accurate samples are included in the final dataset.

## B.3 EXAMPLES OF SYNTHESIZED RIDDLES INSTANCES

Figure 6: Examples of synthesized riddle instances in Numerical, Positional, and Attribute

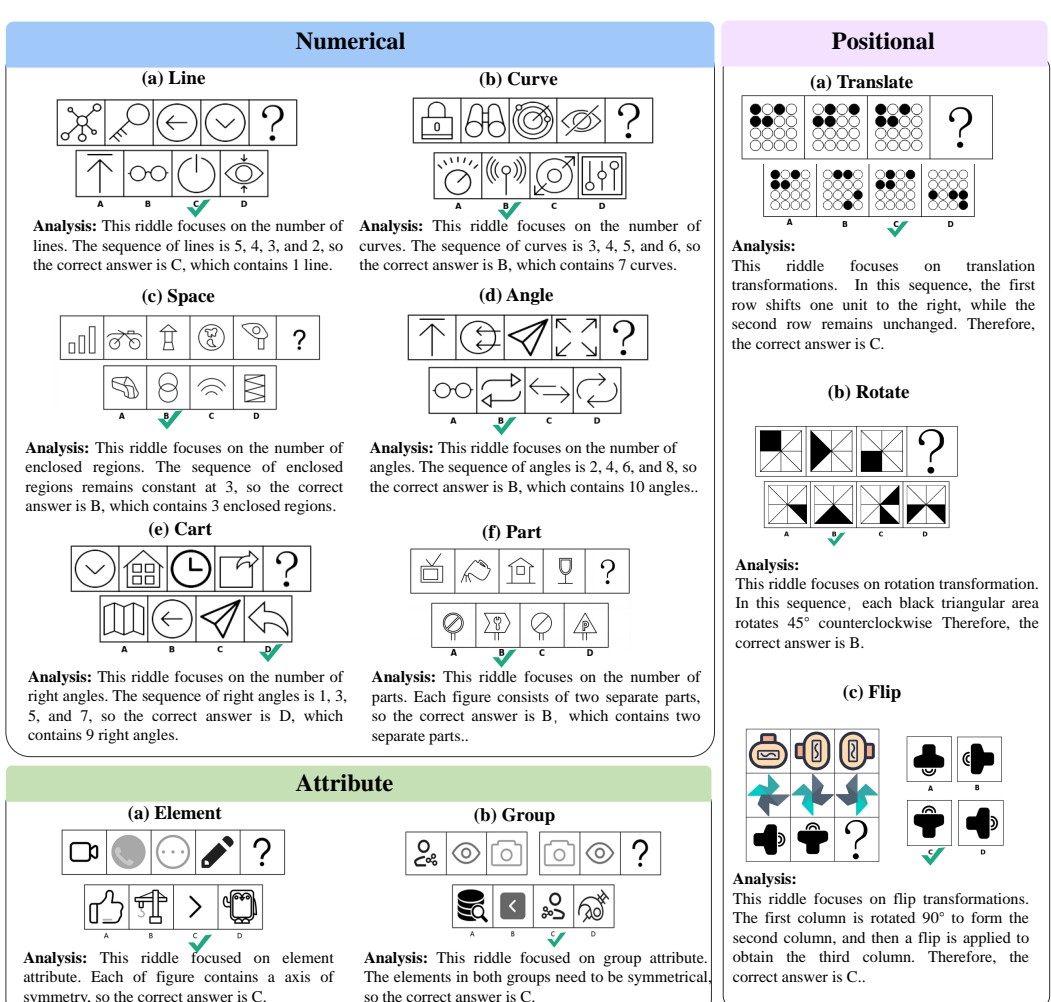

---

Figure 7: Examples of synthesized riddle instances in Spatial, Stylistic, Sudoku, and RAVEN

## Spatial

### (a) Unfolding

**Analysis:** This riddle focuses on the unfolded cube. Option B matches the relative positions of the patterns.

### (b) Three-View

**Analysis:** This riddle focuses on the 3D structure's three views. Option B represents its top view.

### (c) 3D-Reconstruction

**Analysis:** This riddle focuses on reconstructing a 3D structure. Option B's front view and left-side view match the given ones in the problem

### (d) View-Consistency

**Analysis:** This riddle focuses on view-consistency. Option C's side view aligns with the given sequence.

### (e) Multiple-Views

**Analysis:** This riddle focuses on multiple-views. Option A represents different perspectives of the object given in the problem.

## Stylistic

### (a) AND

**Analysis:** This riddle focuses on the *AND* operation of patterns, where *Col. 1 AND Col. 2 = Col. 3.* Therefore, D is the correct answer.

### (b) OR

**Analysis:** This riddle focuses on the *OR* operation of patterns, where *Col. 1 OR Col. 2 = Col. 3.* Therefore, B is the correct answer.

### (c) XOR

**Analysis:** This riddle focuses on the *XOR* operation of patterns, where *Col. 1 XOR Col. 2 = Col. 3.* Therefore, B is the correct answer.

### (d) XNOR

**Analysis:** This riddle focuses on the *XNOR* operation of patterns, where *Col. 1 XNOR Col. 2 = Col. 3.* Therefore, B is the correct answer.

## Sudoku

**Analysis:** The goal is to fill all empty cells so that every row, column, and subgrid contains the digits 1–9 exactly once.

## Raven

**Analysis:** The task is to infer the underlying rules governing the arrangement and select the option that best completes the matrix.

## C    DETAILS OF EVALUATED MODELS

We evaluate seventeen MLLMs, including eleven open-source models and six advanced proprietary models, as shown in Table  6. Human reference answers are obtained from the Fenbi Educational Platform, a widely used source for civil service exam preparation. Notably, GPT-o1, Claude-3-7-Sonnet-Thinking, and Gemini-2.5-Pro are equipped with specialized "thinking" modes aimed at enhancing multi-step reasoning. To further explore reasoning capabilities, CoT prompting is applied to GPT-4o, Qwen2.5-VL-72B, and InternVL2.5-78B-MPO, selected for their reasoning capacity, model scale, or lack of built-in "thinking" mode. Inference for all models is performed using the vLLM framework with unified decoding configurations. The specific evaluation prompts are detailed as follows: the non-CoT variant is shown in Figures 8 and the CoT variants are presented in Figures 9, Figures 10 and Figures 11.

Table 6: Overview of evaluated MLLMs.

| Model Name | Params | Think | CoT | Provider | Version |
|---|---|---|---|---|---|
| *Open-Source MLLMs* | | | | | |
| Qwen2.5-VL-7B Bai et al. (2025) | 7B | ✗ | ✗ | Alibaba | - |
| Qwen2.5-VL-32B | 32B | ✗ | ✗ | Alibaba | - |
| Qwen2.5-VL-72B | 72B | ✗ | ✗ | Alibaba | - |
| Qwen2.5-VL-72B(CoT) | 72B | ✗ | ✓ | Alibaba | - |
| InternVL2.5-8B Chen et al. (2024a) | 8B | ✗ | ✗ | Shanghai AI Lab | - |
| InternVL2.5-MPO-8B Wang et al. (2024b) | 8B | ✗ | ✗ | Shanghai AI Lab | - |
| InternVL2.5-38B | 38B | ✗ | ✗ | Shanghai AI Lab | - |
| InternVL2.5-38B-MPO | 38B | ✗ | ✗ | Shanghai AI Lab | - |
| InternVL2.5-78B | 78B | ✗ | ✗ | Shanghai AI Lab | - |
| InternVL2.5-78B-MPO | 78B | ✗ | ✗ | Shanghai AI Lab | - |
| InternVL2.5-78B-MPO(CoT) | 78B | ✗ | ✓ | Shanghai AI Lab | - |
| DeepSeek-VL2 Wu et al. (2024) | 32B | ✗ | ✗ | DeepSeek | - |
| MiniCPM-V 2.6 Yao et al. (2025) | 8B | ✗ | ✗ | ModelBest | - |
| Dots.vlm1 | 671B | ✗ | ✗ | Rednote hi lab | - |
| *Commercial Products* | | | | | |
| GPT-4o | - | ✗ | ✗ | OpenAI | 2024.11.20 |
| GPT-4o(CoT) | - | ✗ | ✓ | OpenAI | 2024.11.20 |
| o3 | - | ✓ | ✗ | OpenAI | 2025.04.16 |
| Claude-3.7-Sonnet | - | ✗ | ✗ | Anthropic | 2025.02.19 |
| Claude-3.7-Sonnet-Thinking | - | ✓ | ✗ | Anthropic | 2025.02.19 |
| Gemini-2.0-Flash-Thinking | - | ✓ | ✗ | Google DeepMind | 2025.01.21 |
| Gemini-2.5-Flash-Thinking | - | ✓ | ✗ | Google DeepMind | 2025.06.21 |
| Gemini-2.5-Pro | - | ✓ | ✗ | Google DeepMind | 2025.03.25 |
| GPT-5 | - | ✓ | ✗ | OpenAI | 2025.08.07 |

---

**Multiple-choice**

**Task Definition:**
Please select the correct answer based on the question and the provided image, and output the answer.

**Note:**

- The final answer should be provided as follows:
  <answer>X</answer>: The final answer, where X should be one of the answer choices A, B, C, or D.

- The final answer must be in <answer> </answer>.

- <answer>X</answer>: The final answer, where X should be A, B, C, or D.

**Sudoku**

**Task Definition:**
Please solve the following Sudoku puzzle. The basic requirement of Sudoku is to fill each empty space with any Arabic numeral from 1 to 9 such that in each row, each column, and each adjacent 3×3 small square of the same color, the numbers filled in must be from 1 to 9 without any repetition.

**Note:**

- The final answer must be provided in the format <answer>X</answer>, where X is the solved Sudoku grid.

- The solution should be in the form of a 9x9 grid, where each row contains exactly 9 digits (from 1 to 9) with no spaces between them, and rows are separated by a newline character (\n).

- The final answer must be enclosed within <answer> </answer>.

**Raven**

**Task Definition:**
Please solve the following raven puzzle.

**Note:**

- The final answer must be in <answer> </answer>.

- <answer>X</answer>: The final answer, where X should be exactly one of the option labels A–H.

---

Figure 8: Evaluation prompt for responses to various types of riddles.

---

**Prompt for CoT response to multiple-choice riddles**

**Task Definition:**
Please select the correct answer based on the question and the provided image, and output the answer. You must output reasoning process step-by-step.

**Follow these steps to solve the problem:**

1. **Analyze the Image**: Look at the image and identify key details.
2. **Summarize the Pattern**: Find any patterns or relationships in the image.
3. **Evaluate the Options**: Review the answer choices and see which one matches the pattern.
4. **Choose the Best Answer**: Select the answer that best fits the pattern.

**Note:**

- The final answer must be in <answer> </answer>.
- <answer>X</answer>: The final answer, where X should be A, B, C, or D.

---

Figure 9: Evaluation Prompt for CoT Response to Multiple-choice riddles.

---

**Prompt for CoT response to raven riddles**

**Task Definition:**

Please solve the following raven puzzle. You must output reasoning process step-by-step. Follow these steps:

1. Analyze Elements: Look at all the shapes and their characteristics (e.g., size, color, position, quantity).
2. Identify Patterns: Find any relationships or patterns between the shapes (e.g., changes in size, position, or color).
3. Form a Hypothesis: Based on the pattern, guess what the next shape should be.
4. Derive the Answer: Use your hypothesis to find the correct answer, making sure it matches the pattern.

**Note:**

- <answer>X</answer>: The final answer, where X should be exactly one of the option labels A–H.
- the final answer must be in <answer> </answer>

---

Figure 10: Evaluation Prompt for CoT Response to Raven riddles.

**Prompt for CoT response to sukudo riddles**

**Task Definition:**
Please solve the following Sudoku puzzle. The basic requirement of Sudoku is to fill each empty space with any Arabic numeral from 1 to 9 such that in each row, each column, and each adjacent 3×3 small square of the same color, the numbers filled in must be from 1 to 9 without any repetition. You must output reasoning process step-by-step.

Follow these steps:

1. **Analyze the Initial Grid**: Look at the given Sudoku puzzle. Identify which cells are already filled with numbers, and which ones are empty. Also, notice the numbers in each row, column, and 3x3 subgrid.

2. **Identify the Constraints**: Each row, each column, and each 3x3 subgrid must contain the numbers 1 through 9 without repetition. Based on this rule, identify which numbers are missing from the rows, columns, or subgrids.

3. **Start with Obvious Placements**: Look for rows, columns, or subgrids where there is only one possible place for a missing number. Fill these cells in first.

4. **Use Logical Deduction**: If you cannot immediately fill in a number, use logical reasoning to eliminate possibilities. For each empty cell, consider the numbers already present in the same row, column, and 3x3 subgrid, and narrow down the potential candidates for that cell.

5. **Iterate and Fill in More Numbers**: As you fill in more numbers, you will uncover more obvious placements. Continue using logical deduction to fill in the grid.

6. **Complete the Puzzle**: Continue applying these steps until the Sudoku grid is fully solved.

**Note:**

- The final answer must be provided in the format '<answer>X</answer>', where X is the solved Sudoku grid.

- The solution should be in the form of a 9x9 grid, where each row contains exactly 9 digits (from 1 to 9) with no spaces between them, and rows are separated by a newline character ('
n').

- The final answer must be enclosed within '<answer> </answer>'.

Figure 11: Evaluation Prompt for CoT Response to Sudoku riddles.

# D EXPERIMENTAL RESULT ON VISULOGIC

Table 7: **Experimental result on VisuLogic Benchmark.** The best results are marked **bold** and the second results are underlined.

| Models | Param. | Quantity | Spatiality | Position | Attribute | Style | Other | Overall |
|---|---|---|---|---|---|---|---|---|
| Human | - | 45.3 | 52.7 | 71.1 | 50.0 | 47.5 | 44.2 | 51.4 |
| **MLLM Description→LLM** | | | | | | | | |
| Deepseek-R1 | 670B | 27.7 | 23.5 | 24.0 | 27.8 | 23.0 | 35.0 | 26.6 |
| Qwen2.5-72B-Instruct | 72B | 30.2 | 24.4 | 27.5 | 26.5 | 26.8 | 30.8 | 28.0 |
| Claude-3.7-Sonnet (20250219) | - | 26.6 | 22.5 | 25.0 | 28.0 | 25.6 | 30.6 | 25.9 |
| Doubao-1.5-Pro-32k (20250115) | - | 30.0 | 22.5 | 25.0 | 25.6 | 30.0 | 24.1 | 26.6 |
| **Close Source MLLMs** | | | | | | | | |
| GPT-4o-mini (20240718) | - | 27.2 | 23.4 | 23.5 | 18.3 | 31.1 | 16.7 | 24.3 |
| GPT-4o (20240806) | - | 28.6 | 24.7 | 27.2 | 26.8 | 20.0 | 25.9 | 26.3 |
| Kimi-latest | - | 24.9 | 29.4 | 26.5 | 28.0 | 16.7 | 26.9 | 25.9 |
| Doubao-1.5-Vision-Pro-32k (20250115) | - | 28.1 | 23.8 | 29.1 | 25.1 | 32.1 | 35.0 | 28.1 |
| Gemini-2.0-Pro (20250205) | - | 29.7 | 24.2 | 27.9 | 30.5 | 22.2 | 33.3 | 28.0 |
| Claude-3.7-Sonnet (20250219) | - | 22.7 | 27.3 | 27.9 | 28.0 | 22.2 | 22.2 | 24.8 |
| **Open Source MLLMs** | | | | | | | | |
| LLaVA-v1.5 | 7B | 26.1 | 24.2 | 23.5 | 17.1 | 31.1 | 22.2 | 24.6 |
| LLaVA-OneVision (SI) | 7B | 22.4 | 27.3 | 33.1 | 23.2 | 25.6 | 22.2 | 25.3 |
| Ovis2 | 8B | 26.1 | 23.8 | 27.2 | 28.0 | 25.6 | 24.1 | 25.6 |
| Qwen2.5-VL-7B-Instruct | 7B | 27.6 | 20.9 | 25.2 | 23.2 | **37.8** | 25.0 | 26.0 |
| Qwen2.5VL-72B-Instruct | 72B | 25.2 | 23.8 | 27.2 | 25.6 | 25.6 | 34.3 | 26.2 |
| InternVL2.5-38B | 8B | 24.4 | 26.4 | 27.2 | 23.2 | 25.6 | 26.9 | 25.5 |
| InternVL2.5-78B | 78B | 26.6 | 26.0 | 26.5 | 26.8 | 31.1 | 30.6 | 27.3 |
| InternVL3-38B | 38B | 28.7 | 27.6 | 26.1 | 21.4 | 23.9 | 28.5 | 27.1 |
| InternVL3-78B | 78B | 27.7 | 26.1 | **31.6** | 26.3 | 21.3 | 32.3 | 27.7 |
| Qwen2.5-VL-7B-Instruct-SFT (VisuLogic) | 7B | 24.4 | 26.4 | 27.2 | 23.2 | 25.6 | 26.9 | 25.5 |
| Qwen2.5-VL-7B-Instruct-RL (VisuLogic) | 7B | 26.6 | **33.8** | 29.4 | 23.2 | 18.9 | 29.6 | 28.0 |
| InternVL2.5-38B-SFT (VisuLogic) | 38B | 30.6 | 29.4 | 20.6 | 25.6 | 30.0 | 25.0 | 27.9 |
| InternVL2.5-38B-RL (VisuLogic) | 38B | **31.2** | 31.2 | 26.5 | 30.5 | 30.0 | **38.9** | **31.1** |
| **Ours** | | | | | | | | |
| PAVR | 7B | 28.9 | 30.3 | **31.6** | **34.1** | 30.0 | 33.3 | 30.6 |

# E USAGE OF LARGE LANGUAGE MODELS

We thank DeepSeek for assisting in polishing the language and improving the overall clarity of this manuscript.

