# OpenReview forum: "VisuRiddles: Fine-grained Perception is a Primary Bottleneck for Multimodal Large Language Models in Abstract Visual Reasoning"
_ICLR.cc/2026/Conference — ICLR 2026 Poster_

### Official Review · Reviewer_VP3M · 2025-10-20

**Soundness:** 3
**Presentation:** 3
**Contribution:** 3
**Rating:** 6
**Confidence:** 3

**Summary:**

Current Multimodal Large Language Models (MLLMs) still exhibit significant limitations in Abstract Visual Reasoning (AVR) tasks. To address this issue, this paper introduces the VisuRiddles benchmark. To tackle the dual challenges of perception and reasoning inherent in AVR, the authors develop the Perception-Augmented Visual Reasoner (PAVR), which integrates enhanced perception capabilities with refined reasoning abilities. Through a novel SFT+RL paradigm, the model's fine-grained perception for AVR tasks is substantially improved, and the effectiveness of this approach is demonstrated through rigorous experiments.

**Strengths:**

1. The structure of this paper is extremely clear, presenting a definite logical flow.
2. The paper thoroughly discusses the deficiencies of existing general-purpose MLLMs in Abstract Visual Reasoning and validates this observation by introducing the VisuRiddles Benchmark, a comprehensive testbed covering multiple reasoning dimensions.
3. It innovatively proposes the VisuRiddles Synthesizer, an ingenious data synthesizer that generates a diverse set of AVR instances with detailed perception annotations. These instances are highly interpretable and reusable, making this a valuable contribution for future research aimed at enhancing the AVR capabilities of MLLMs.
4. The paper employs a systematic two-stage training paradigm (SFT+RL) that effectively enhances the model's abstract visual reasoning capabilities.
5. The experimental section comprehensively presents the performance of current open-source and closed-source MLLMs on the VisuRiddles Benchmark, showcasing the remarkable performance achieved by the PAVR model after the two-stage training. Furthermore, well-designed ablation studies are conducted to demonstrate the rationale and contribution of each component in the training process.

**Weaknesses:**

1. Detailed experimental settings and configurations should be included in the appendix to ensure the reproducibility of the reported results.
2.  A central claim of the paper is that deficient fine-grained perception is the primary bottleneck for MLLMs on AVR tasks. While the experiment in Table 3 supports this, it also shows that commercial MLLMs achieve only around 30% accuracy on Raven-like tasks even when provided with perception descriptions ('P') as input. This raises a question: is this poor performance due to the perception descriptions still being insufficiently comprehensive, or is it due to the inherent limitations of the models' reasoning capabilities? Further experiments are required to more conclusively disentangle these factors.
3. The trained model should also be evaluated on a suite of standard MLLM benchmarks. This would verify that the model's original, general-purpose capabilities have not been compromised by the specialized training (i.e., to check for catastrophic forgetting).
4. The paper states that common reasoning-enhancement strategies, such as increasing model parameter scale or applying Chain-of-Thought (CoT) prompting, yield limited benefits on AVR tasks. However, it does not provide sufficient detail on the specific implementation of the CoT prompts used for evaluating the baseline models.
5. The experiments are conducted exclusively on the Qwen-VL-7B model. To validate the generalizability of the proposed training paradigm, it should be tested on a wider range of MLLMs with varying architectures and parameter scales.
6. The paper would be more persuasive，if it could be directly compared with other MLLMs specifically designed to enhance reasoning abilities.

**Questions:**

1. The patterns in VisuRiddles are relatively simple and clean geometric shapes. If these shapes are replaced with more complex or "noisy" objects (such as small icons of real-world objects like cats or cars, or complex fractal patterns), will the model fail in the abstract reasoning task due to being disturbed by the semantic content of the complex objects? For a given abstract rule (such as "the number of elements increases one by one"), if the elements undergo a simple style change (for example, from black squares to blue triangles), will the model's accuracy be affected? This will test to what extent the model generalizes the abstract rule itself rather than merely memorizing specific visual features.
2. The paper convincingly demonstrates that fine-grained perception is the key bottleneck. However, Table 3 shows that even when perception descriptions (denoted as "P") are provided, powerful models like GPT-4V still perform poorly (with an accuracy rate of approximately 30%). What is the essence of these failures? When perception prompts are provided, do the models fail to understand the prompts, or do they understand the prompts but make fundamental logical errors? If a better perception description is provided, will the models be able to correctly arrive at the answer? If not, does this prove that reasoning ability remains the core limitation?
3. The SFT+RL mentioned in this article is a training paradigm specifically for the VisuRiddles domain. Has the final PAVR model been evaluated on standard and general multi-modal large model benchmark tests (such as MMBench, etc.)? Will such training weaken the original basic capabilities of the model?
4. The experiments in the paper were conducted only on the Qwen2.5-VL-7B model. Have you ever attempted to apply this SFT+RL training paradigm to other models with different parameter sizes or to other MLLM architectures?
5. The paper states that the benefits brought by the Chain-of-Thought (CoT) hinting mechanism to the baseline model are limited. Could you provide the detailed information for this part?
6. There are many MLLM models that enhance reasoning capabilities. They can also be compared with PAVR on the VisuRiddles benchmark.

---

> ### Author Response · Authors · 2025-11-24
> **Response to Reviewer VP3M [1 of 2]**
>
> Thanks for constructive feedback and appreciation of our work.
>
> $\textbf{Response to W1:}$
>
> We sincerely thank the reviewer for this constructive suggestion. In the revised manuscript, we have supplemented comprehensive experimental settings, including specific hyperparameters for both training and evaluation, in Section 5.1 and Appendix C. The revised paragraph now includes the training details:
>
> >We use Qwen2.5-VL-7B as the baseline model to develop PAVR. In the supervised fine-tuning stage, the model is trained for 20 epochs using 20K synthesized AVR instances generated by the VisuRiddles Synthesizer.
> >For this stage, we employed the AdamW Optimizer, with a Batch Size of 16, and a Learning Rate set at $5\mathrm{e}{-7}$. Subsequently, in the GRPO stage, optimization is performed using 4K synthesized instances over 40 epochs.
> >For this stage, the Learning Rate was set to $1\mathrm{e}{-6}$, the Rollout Number was 5, and the KL Loss Coefficient and CLIP Ratio were 0.01 and 1.0, respectively. All experiments are conducted on a setup comprising 8 NVIDIA A800 80G GPUs. Evaluation settings are summarized in Appendix C.
>
> and evaluation prompts for various types of riddles. We have shown one of the examples as follows:
>
> > **Task Definition:**
> > Please solve the following Sudoku puzzle. The basic requirement of Sudoku is to fill each empty space with any Arabic numeral from 1 to 9 such that in each row, each column, and each adjacent 3×3 small square of the same color, the numbers filled in must be from 1 to 9 without any repetition.
> >
> > **Note:**
> > - The final answer must be provided in the format `<answer>X</answer>`, where X is the solved Sudoku grid.
> > - The solution should be in the form of a 9x9 grid, where each row contains exactly 9 digits (from 1 to 9) with no spaces between them, and rows are separated by a newline character (`\n`).
> > - The final answer must be enclosed within `<answer> </answer>`.
> >
> > **Raven**
> >
> > **Task Definition:**
> > Please solve the following raven puzzle.
> >
> > **Note:**
> > - The final answer must be in `<answer> </answer>`.
> > - `<answer>X</answer>`: The final answer, where X should be exactly one of the option labels A–H.
> >
> > _Evaluation prompt for responses to various types of riddles._
>
> More details are provided in Appendix C.
>
> $\textbf{Response to W2 and Q2:}$
>
> Thank you for raising this valuable question. The results in Table 3 demonstrate that eliminating perceptual bias solves the majority of AVR problems. However, performance on high-level tasks like RAVEN remains limited even with accurate descriptions. This is primarily because the abundance of visual elements in RAVEN puzzles results in extremely long and complex descriptions, which challenges the models' long-context reasoning capabilities. We also encountered this issue during data synthesis, where such complexity reduced our data utilization rate. We plan to investigate this specific challenge further in future research.
>
> $\textbf{Response to W4:}$
>
> We sincerely thank the reviewer for pointing out this omission. We agree that providing specific details on the CoT implementation is essential for reproducibility. In the revised manuscript, we have included the CoT prompts used for evaluating the baseline models in Appendix C. We present a representative example:
>
> > **Prompt for CoT response to multiple-choice riddles**
> >
> > **Task Definition:**
> > Please select the correct answer based on the question and the provided image, and output the answer. You must output reasoning process step-by-step.
> >
> > **Follow these steps to solve the problem:**
> > 1. **Analyze the Image:** Look at the image and identify key details.
> > 2. **Summarize the Pattern:** Find any patterns or relationships in the image.
> > 3. **Evaluate the Options:** Review the answer choices and see which one matches the pattern.
> > 4. **Choose the Best Answer:** Select the answer that best fits the pattern.
> >
> > **Note:**
> > - The final answer must be in `<answer> </answer>`.
> > - `<answer>X</answer>`: The final answer, where X should be A, B, C, or D.
> >
> > _Evaluation Prompt for CoT Response to Multiple-choice riddles._
>
> More details are provided in Appendix C of the revised manuscript.

---

> ### Author Response · Authors · 2025-11-24
> **Response to Reviewer VP3M [2 of 2]**
>
> $\textbf{Response to W3 and Q3:}$
>
> We thank the reviewer for raising the concern regarding the loss of the model’s general capabilities. This is recognized as a core risk inherent to specialized training. However, it should be clarified that our core contribution lies in verifying that perception is the primary bottleneck for MLLMs and successfully tackling the AVR problem.
>
> In practice, the gold standard solution for addressing catastrophic forgetting is Data Blending. During specialized fine-tuning stage, the specialized data (VisuRiddles instances) should be blended with the model's original general SFT data. However, since the general finetune dataset for Qwen2.5VL has not been publicly released, we are unable to apply this solution to the baseline to test and prevent forgetting.
>
> Despite this constraint, we conducted experimental exploration using another larger-scale internal commercial business model. After blending the VisuRiddles synthetic data with the general finetune data, the model's performance not only stabilized but also saw improvement across all domains:
>
> | Train Data | InfoVQA[1] | MathVision[2] | MMMU pro[3] | Marvel (AVR)[4] |
> | :--- | :---: | :---: | :---: | :---: |
> | General SFT Data | 82.4 | 33.2 | 45.3 | 30.0 |
> | Blended Data (General + VisuRiddles) | 83.3 (+0.9) | 35.0 (+1.8) | 47.8 (+2.5) | 34.2 (+4.2) |
>
> Therefore, we believe any potential decline in general capabilities on Qwen is due to the technical limitation of lacking data blending, rather than a flaw in the PAVR training paradigm itself.
>
> [1] Mathew M, Bagal V, Tito R, et al. Infographicvqa[C]//Proceedings of the IEEE/CVF Winter Conference on Applications of Computer Vision. 2022: 1697-1706.
>
> [2] Awais M, Ahmed T, Aslam M, et al. Mathvision: An accessible intelligent agent for visually impaired people to understand mathematical equations[J]. IEEE Access, 2024.
>
> [3] Yue X, Zheng T, Ni Y, et al. Mmmu-pro: A more robust multi-discipline multimodal understanding benchmark[C]//Proceedings of the 63rd Annual Meeting of the Association for Computational Linguistics (Volume 1: Long Papers). 2025: 15134-15186.
>
> [4] Jiang Y, Zhang J, Sun K, et al. Marvel: Multidimensional abstraction and reasoning through visual evaluation and learning[J]. Advances in Neural Information Processing Systems, 2024, 37: 46567-46592.
>
> $\textbf{Response to W5 and Q4:}$
>
> We thank the reviewer for their attention to this critical question. We believe the current experiments, anchored by the Qwen-2.5-vl baseline, are sufficient to successfully validate the efficacy of our proposed Perception-Augmented training paradigm.
> However, We wish to clarify that while fine-tuning with both the SFT and GRPO stages is highly resource-intensive and time-consuming, we have already begun testing generalization on alternative models to address the reviewer's concern.
>
> We are confident that the PAVR paradigm is universal and that the experimental results will align our baseline. We believe that expanding architectural testing is a valuable link or direction for future research.
>
> $\textbf{Response to W6 and Q6:}$
>
> Our current evaluation includes the following models:
>
> 1) Commercial Models with Native Thinking Modes (gemini 2.5 flash thinking, gemini2.5-pro, gpt-5, o3).
>
> 2) Open-Source and Inference Enhanced Models (InternVL-2.5-MPO and qwen2.5VL-CoT).
>
> To ensure our comparison remains at the frontier, we have supplemented our results with the latest SOTA open-source models: Qwen3-VL-235B-Thinking and Qwen3-VL-235B-Instruct in Table 2 of the revised manuscript.
>
> These models represent the contemporary landscape across various reasoning paradigms. If there are any other specific, we would be glad to supplement the results.
>
> $\textbf{Response to Q1:}$
>
> We strongly agree with this view. Whether introducing more complex or 'noisy' objects or implementing simple style changes, the model's accuracy would be negatively affected, potentially leading to failure in AVR tasks. We are committed to continuously optimizing this issue.
>
> $\textbf{Response to Q5:}$
>
> The experimental results in Table 2 show that the performance improvement brought by using CoT is minimal, and even degrades in some cases. The reason for this phenomenon is shown in Figure 4. The failure of CoT is due to perceptual errors directly misleading the selection of the answer.

---

### Official Review · Reviewer_FgFP · 2025-10-25

**Soundness:** 2
**Presentation:** 3
**Contribution:** 2
**Rating:** 2
**Confidence:** 5

**Summary:**

This paper identifies fine-grained visual perception as a key bottleneck limiting the reasoning abilities of multimodal large language models (MLLMs). To study this limitation, the authors introduce VisuRiddles, a benchmark designed to assess Abstract Visual Reasoning (AVR) across multiple dimensions such as numerosity, attributeness, style, position, and spatiality. They also propose a new method, Perception-Augmented Visual Reasoner (PAVR), which applies supervised fine-tuning (SFT) and reinforcement learning (GRPO) to train Qwen2.5-VL-7B on VisuRiddles. Empirical results show that PAVR outperforms several zero-shot MLLMs, suggesting that perception-oriented fine-tuning can partially mitigate AVR limitations.

**Strengths:**

1. The paper tackles an important and timely problem - the limited fine-grained perception of MLLMs - which is of clear relevance to the ICLR community.
2. Introduction of a new benchmark (VisuRiddles) focused specifically on AVR is valuable.
3. The benchmark covers diverse aspects of perceptual reasoning (numerosity, attributeness, style, position, spatiality).
4. The paper demonstrates clear limitations of several open and proprietary model families on the benchmark.
5. The proposed method achieves improved results over zero-shot baselines, showing the benefit of perception-focused training.
6. The literature review is comprehensive and the writing is generally clear and accessible.

**Weaknesses:**

1. Limited novelty of the benchmark: The benchmark largely repackages existing datasets (Chinese National Civil Service Examination, RAVEN, and Sudoku) with limited methodological innovation. Details of dataset construction are missing. For instance:
    1. Were questions from the Civil Service dataset used as-is or modified?
    2. How were 100 RAVEN questions selected from the 70k instances?
    3. A pseudo-code description of the “Synthesis Algorithm” (Fig. 2a) would improve reproducibility.
2. Limited methodological novelty: The approach (SFT + GRPO on Qwen2.5-VL-7B) applies standard fine-tuning methods rather than introducing new learning techniques. The contribution is mainly empirical.
3. Unfair evaluation setup: The trained PAVR model is evaluated on the same benchmark it was fine-tuned on, whereas competing models are evaluated zero-shot. This limits the fairness and interpretability of the comparison.
4. Missing training details: The paper does not specify the train/validation/test splits used for fine-tuning PAVR, leaving open the possibility of data leakage or overfitting to the benchmark distribution. The claimed improvements might be in-distribution effects rather than true reasoning gains.
5. Limited analysis: While Appendix D attempts to evaluate transfer to other tasks, details are insufficient. Stronger evidence is needed to show that perceptual training benefits general reasoning.
6. The discussion stops short of exploring why models fail across AVR dimensions. A qualitative error analysis would be helpful.
7. The structured perceptual descriptions and reasoning chains are generated using only Gemini 2.5-Flash-Think. The paper should justify this choice and assess how dependent results are on this specific model.
8. Missing baselines: The evaluation omits several relevant MLLMs (e.g., LLaVA, InstructBLIP) and non-MLLM baselines such as supervised CNNs or few-shot methods (e.g., Prototypical Networks, SNAIL, MetaBaseline).
9. Figure 1a shows some models performing below random-guess levels without discussion or explanation.



Minor comments:
1. Describing VisuRiddles as “real-world data” is misleading since the benchmark primarily consists of synthetic, 2D, geometry-based tests.
2. Clarify the distinction between Position and Spatiality dimensions.
3. Specify whether few-shot evaluation was used for baseline models.
4. The description of prior benchmark limitations in Section 2.1 (“partial reliance on external knowledge and lack breadth in reasoning coverage”) is too vague—consider providing examples.
5. Section 2.2 should explicitly highlight how PAVR differs from prior approaches.
6. The claim “we target the most challenging part of AVR tasks” (Section 3.1) is unsupported—please justify what makes these dimensions the most difficult.
7. Clarify how “question length” is computed in Table 1 for image-based data, given that tokenization methods differ across models.

**Questions:**

1. How exactly was the data from each source processed and filtered? Could you include pseudo-code or a more detailed benchmark construction pipeline?
2. How are the train, validation, and test splits organized for PAVR training?
3. Why was Gemini 2.5-Flash-Think used for generating reasoning chains, and how sensitive is the system to this model choice?
4. Did you consider evaluating on additional MLLMs (e.g., LLaVA, InstructBLIP) or on supervised/few-shot models?
5. How do you interpret the below-random results observed in Fig. 1a?

**Details Of Ethics Concerns:**

Can problems from Chinese National Civil Service Examination be incorporated in the released benchmark?

---

> ### Comment · Reviewer_emm3 · 2025-11-13
> **Regarding the copyright issue of civil service exam questions**
>
> Chinese national here. The civil service exam questions from previous years are widely shared at the source website Fenbi, where the authors collected for VisuRiddle: https://spa.fenbi.com/cube-module-cms/ .
>
> This is similar to Gaokao, aka the college entrance exams, used in existing datasets like this one: https://arxiv.org/abs/2305.12474 .

---

> ### Author Response · Authors · 2025-11-24
> **Response to Reviewer FgFP [1 of 3]**
>
> We sincerely thank the reviewer for their unique perspective offered on potential alternative methodologies.
>
> $\textbf{Response to W1 and M1:}$
>
> We appreciate the attention to our data. Our benchmark data was not collected directly from the synthesis framework or the Chinese National Civil Service Examination. Instead, it underwent a rigorous “Collection, Cleaning, and Consolidation” pipeline, as detailed in Section 3.1 and Appendix A.2. This process was essential to introduce this high-quality real-world human cognitive assessment data into the MLLM community.
>
> $\textbf{Response to W2:}$
>
> Our core contribution does not lie in proposing novel optimization algorithms or improving existing models, but in being the first to propose and verify that Perception-Augmented is the key path to solving AVR tasks. Crucially, to solve this problem, we introduced the VisuRiddles Synthesizer to generate structured perceptual descriptions, enabling the model to "translate" abstract graphics into precise language, a feat unattainable through standard data fine-tuning.
>
> Furthermore, building upon this acquired perceptual capability, RL is utilized to address the instability in selecting perceptual strategies. Ultimately, the novelty of PAVR lies in providing an end-to-end solution targeted at the AVR bottleneck, constructing the perceptual foundation via synthetic data, and utilizing RL to achieve a reasoning performance leap.
>
> $\textbf{Response to W3:}$
>
> It is necessary to first clarify that PAVR was not fine-tuned on the VisuRiddles benchmark itself. PAVR was trained only on the 20K (SFT) and 4K (RL) synthetic instances generated by the VisuRiddles Synthesizer to enhance perceptual ability and strategy optimization.
>
> Evaluation was conducted on the VisuRiddles Benchmark, whose core source is real-world Chinese Civil Service Examinations. Therefore, all models were evaluated on the final Benchmark in a strictly zero-shot setting. In fact, while advanced closed-source models (e.g., GPT-4o) and open-source models (e.g., Qwen2.5VL) are trained on massive image-pair data, they still perform poorly on AVR tasks. PAVR’s significant performance leap, achieved after enhancement with only 24K synthetic data, verified a more challenging hypothesis: whether foundational capability learned from synthetic data can successfully generalize to solve unseen, complex, real-world problems.
>
> $\textbf{Response to W4:}$
>
> VisuRiddles consists of two parts: the VisuRiddles Benchmark, which is primarily collected from real question types for evaluation, and the VisuRiddles Synthesizer, which uses source code to automatically generate AVR instances with structured perceptual descriptions for training. We perform evaluation on the 1K real instances of the former, and train on the 24K synthetic instances generated by the latter.
>
> The samples for training and testing do not overlap, therefore there is no risk of data leakage. Our results are not the in-distribution effects the reviewer worries about, but rather the OOD generalization performance from synthetic instances to real-world reasoning tasks. Furthermore, during the training process, we similarly use additional synthetic data as a validation set to monitor convergence. Therefore, the objectivity and fidelity of the evaluation results can be guaranteed.
>
> $\textbf{Response to W5:}$
>
> Our experiments in Table 3 demonstrate that once the input is replaced from the raw image (V) with perfect perceptual descriptions (P), the model's average accuracy achieves a substantial boost. This proves that the model itself possesses the potential for solving complex logic problems. Our perceptual training is precisely the method to eliminate the visual bottleneck, which is a fundamental issue, thereby enabling the model to successfully complete AVR tasks.
>
> $\textbf{Response to W6:}$
>
> The primary failure modes observed across the seven dimensions are as follows:
>
> Numerosity: The model fails to precisely count the number of lines, points, or shapes.
>
> Attribute: The model suffers from feature hallucination and misidentification.
>
> Position: The model exhibits errors in tracking the position of elements.
>
> Stylistic: The model mistakes one pattern for another.
>
> Spatiality: Failure in three-dimensional modeling.
>
> Sudoku: Errors in logical deduction.
>
> RAVEN: Information overload and failure to identify abstract rules.

---

> ### Author Response · Authors · 2025-11-24
> **Response to Reviewer FgFP [2 of 3]**
>
> $\textbf{Response to W7 and Q3:}$
>
> We should first clarify that the structured perceptual descriptions are programmatically composed from the underlying metadata during the synthesis process, rather than being generated by the API. The API is utilized only for generating the CoT data based on the descriptions. This ensures the descriptions are inherently accurate and consistent with the abstract graphics. We selected Gemini 2.5-Flash-Think due to a favorable cost-efficiency trade-off.  it demonstrated comparable CoT generation quality to the Gemini 2.5 Pro model while incurring significantly lower computational costs, making it the most resource-efficient choice for large-scale data synthesis.
>
> $\textbf{Response to W8 and Q4:}$
>
> Regarding non-MLLM baselines, the task proposed in this paper is primarily generative abstract visual reasoning, not a discriminative classification task. The model must input an image and generate a natural language reasoning chain and the final answer. The baselines, including Prototypical Networks, SNAIL, MetaBaseline, are architecturally designed to only output discrete class labels and fundamentally lack text generation and visual reasoning capabilities. We do not know how to complete the evaluation.
>
> As for LLaVA and InstructBLIP baselines, their visual encoder resolution and reasoning capabilities are significantly weaker than the baselines we have already evaluated (e.g., Qwen2.5VL, InternVL-2.5). Introducing these older models provides no new insights when SOTA models are already struggling. To explore the true reasoning limit of MLLMs, we have supplemented with the latest advanced models, Qwen3-VL-235B-Thinking and Qwen3-VL-235B-Instruct, with results presented in Table 2 of the revised manuscript.
>
> $\textbf{Response to W9 and Q5:}$
>
> Thank you for this valuable question. Theoretically, random selection in Figure 1(a) should be close to 25%. Upon our inspection, the reasons why model accuracy falls below the random baseline of 25% include the following two points:
>
> 1) The model is not engaging in random guessing, but rather exhibits high confidence in its erroneous perception, leading it to systematically avoid the correct answer.
>
> 2) As generative models, MLLMs fail if their output format cannot be strictly parsed or if they trigger a refusal mechanism. These instances are counted as zero, which pulls the overall accuracy below the random baseline.
>
> $\textbf{Response to M2:}$
>
> The core distinction between Positional Reasoning and Spatiality Reasoning lies in dimensionality (2D vs. 3D) and the nature of the transformation (planar transformation vs. stereoscopic mapping). Representative examples of these categories are shown in Appendix A.1.
>
> $\textbf{Response to M3:}$
>
> Thank you for raising this question. We used few-shot evaluation for the baseline models.
>
> $\textbf{Response to M4:}$
>
> he claim that prior benchmarks rely on external knowledge and lack breadth can be exemplified as follows: Benchmarks such as MathVerse, MV-MATH, and MMMU often require the model to retrieve inherent domain knowledge (e.g., mathematical theorems or physical laws) that is not visually present in the input, thereby testing knowledge application rather than pure visual logical deduction. Conversely, existing logic-oriented benchmarks like CVR and RAVEN tend to offer a limited breadth of evaluation dimensions, primarily focusing on $3 \times 3$ matrix completion or basic attribute recognition.
>
> $\textbf{Response to M5:}$
>
> Prior work often focused on scaling model parameters, using CoT to boost logical ability, or employing RL to enhance model reasoning. Those methods lead to "black-box inference," where it is unclear if the model truly understands the visual features. We first identified and verified that fine-grained perception is the core bottleneck for AVR tasks. To address this, we proposed the VisuRiddles Synthesizer, using generated structured perceptual descriptions to break the traditional "black-box" QA training, thereby constructing explicit, explainable Perception-Augmented training. Furthermore, we utilize RL to stabilize the perceptual strategy selection and further strengthen the model's overall reasoning ability. Our work offers a reliable path to solving AVR tasks.
>
> $\textbf{Response to M6:}$
>
> The assertion that we target the most challenging parts of AVR tasks is justified by both the inherent cognitive complexity of the dimensions and the empirical failure rate of current state-of-the-art MLLMs. Requirement for Fine-Grained Perception: The five core dimensions, such as Numerosity, Attribute, Style, are challenging because they require MLLMs to perceive subtle visual structures, which is the exact capability often overlooked and lacking in existing models. The absence of this fine-grained perception is the core bottleneck we identify.
>
> $\textbf{Response to M7:}$
>
> The question length statistics are calculated by counting the average number of words contained in each question.

---

> ### Author Response · Authors · 2025-11-24
> **Response to Reviewer FgFP [3 of 3]**
>
> $\textbf{Response to Q1:}$
>
> The automated process for generating training instances is described in Appendix B.2 (Rule Selection, Element Configuration, Rendering). We will also show the pseudo-code for the implementation below:
> > _Pseudo-code for VisuRiddles Synthesizer:_
> >
> > **Phase 1: Rule Selection**
> >
> > 1: RULE ← SELECT_RULE(R)
> >
> > 2: SUB_RULE ← SELECT_SUB_RULE(RULE)
> >
> > **Phase 2: Layout & Element Setup**
> >
> > 3: LAYOUT ← SELECT_LAYOUT(Predefined Layouts)
> >
> > 4: ICONS ← SELECT_OR_GENERATE(Z, SUB_RULE)
> >
> > 5: PARAMS ← GENERATE_RULE_PARAMS(SUB_RULE, LAYOUT)
> >
> > **Phase 3: Instance Construction (Rendering & Labeling)**
> >
> > 6: IMAGE ← VISUAL_RENDER(LAYOUT, ICONS, PARAMS)
> >
> > 7: OPTIONS ← GENERATE_OPTIONS(IMAGE, PARAMS)
> >
> > 8: GT ← COMPUTE_GT(PARAMS)
> >
> > 9: DESC ← GENERATE_PERCEPTUAL_DESC(LAYOUT, ICONS, PARAMS)
> >
> > **Phase 4: API Labeling and Filtering**
> >
> > 10: Input: DataList ← (IMAGE, DESC, GT)
> >
> > 11: Output: Dspr ← (Filtered SFT Data)
> >
> > 12: PROMPT ← FORMAT(Question, DESC, OPTIONS)
> >
> > 13: (PRED, CoT) ← QUERY_GEMINI(PROMPT)  // API Labeling
> >
> > 14: if PRED == GT then;
> >
> > 15: SAMPLE ← (IMAGE, PROMPT, CoT, GT)
> >
> > 16: Dspr.ADD(SAMPLE)  // Filtering/Validation Step
> >
> >17: end if
> >
> >18: return Dspr
>
> $\textbf{Response to Q2:}$
>
> The core VisuRiddles Benchmark (real-world exams) is used entirely as the final test set. And the training and validation splits are organized solely by splitting the synthetic data generated by the VisuRiddles Synthesizer, ensuring the test and training samples remain completely disjoint.

---

### Official Review · Reviewer_TJrV · 2025-10-26

**Soundness:** 3
**Presentation:** 3
**Contribution:** 3
**Rating:** 6
**Confidence:** 3

**Summary:**

This paper investigates the limitations of Multimodal Large Language Models (MLLMs) in Abstract Visual Reasoning (AVR) which are tasks such as Raven’s Progressive Matrices and visual riddles that require structural and relational perception. The authors hypothesize that fine-grained perception, rather than reasoning itself, is the main bottleneck in these models.
To address this, they propose VisuRiddles, which consists of
1) A benchmark of 1000 diverse AVR problems collected and curated from real-world riddles, covering categories such as numerical, positional, stylistic, and spatial reasoning.
2) A synthesizer, which programmatically generates abstract visual puzzles paired with structured perceptual descriptions and chain-of-thought reasoning traces.

With VisuRiddles, they rain a model called the Perception-Augmented Visual Reasoner (PAVR) through a two-stage process:
1) Supervised Fine-Tuning to strengthen fine-grained perceptual understanding using synthetic perceptual annotations.
2) Reinforcement Learning via Group Relative Policy Optimization (GRPO) to improve reasoning reliability and perceptual grounding.

Experiments on the VisuRiddles benchmark show that PAVR significantly outperforms open-source and commercial MLLMs (including GPT-4o, Gemini 2.5, Claude 3.7, and Qwen2.5VL), achieving 46.8% accuracy compared to ~25–33% for top baselines.

**Strengths:**

1) The paper presents rigorous empirical analysis, including comprehensive comparisons with both open-source and commercial MLLMs. Ablation studies are detailed and support the key claims (e.g., perception annotations yield +42.7% improvement, RL adds +7.3% further gains).
2) The findings have broad implications for MLLM development, suggesting that improving perceptual resolution and grounding may be more impactful than simply enlarging models or adding reasoning traces.

**Weaknesses:**

1) While the synthesizer is innovative, the paper admits that generated riddles are “deliberately easier” and may lack the richness and noise of real-world abstract reasoning tasks. This could limit transferability.
2) Although VisuRiddles is comprehensive, stronger evidence of generalization would come from evaluating PAVR on unseen external datasets (e.g., MARVEL, PuzzleVQA, or VisLogic)
3) The “rethinking” phenomenon observed in PAVR is intriguing but underexplored. Quantitative metrics on perceptual correction or reasoning consistency would strengthen the argument.

**Questions:**

1) What measures ensure that the perceptual descriptions generated by the API labeling stage are accurate and consistent? Could noisy annotations harm training?
2) Is there quantitative evidence that PAVR’s improvements are more from perceptual enhancements than reasoning policy optimization (e.g., via perceptual accuracy metrics)?

---

> ### Author Response · Authors · 2025-11-24
> **Response to Reviewer TJrV [1 of 2]**
>
> We sincerely thank the reviewer for their careful reading and valuable suggestions.
>
> $\textbf{Response to W1:}$
>
> Thanks for this valuable suggestion. We fully agree that increasing richness and noise is an effective strategy to enhance model robustness and transferability. In this work, our primary objective was to strictly verify the core hypothesis that "Fine-grained Perception is a Primary Bottleneck for MLLMs". To isolate this factor, we prioritized clear perceptual signals in the current version. However, we plan to further refine the dataset in future work. Thanks to the modular design of our VisuRiddles Synthesizer, it fully supports lifting the data complexity and realism:
>
> 1）Richness: The framework can easily be expanded with a larger Icon Gallery, diverse Layout templates, and more complex compound logical rules to increase diversity.
>
> 2）Noise: We can introduce various noise types during the rendering phase, including background noise, such as Gaussian noise, or affine transformation noise, such as distortion, rotation and scaling to simulate real-world imperfections.
> Thank you again for pointing out this important direction.
>
> $\textbf{Response to W2:}$
>
> Thank you for raising this question regarding the generalization of AVR tasks. However, directly validating PAVR's generalizability on AVR benchmarks with vastly different sample domains is challenging. We should first clarify that our core contribution lies in verifying that perception is the primary bottleneck for MLLMs and successfully tackling the AVR problem within similar domains (i.e., similar simple rules and their combinations). Even within AVR tasks, different benchmarks exhibit significant discrepancies in visual style and inherent rules. We provide a performance comparison between the Baseline and PAVR on PuzzleVQA, which has a large domain discrepancy with data generated by the VisuRiddles Synthesizer, and VisuLogic (provided in Appendix D), which has a similar domain, as shown in the table below:
>
> | Model | VisuLogic | PuzzleVQA |
> | :--- | :---: | :---: |
> | Baseline | 26.0 | 47.2 |
> | PAVR | 30.6 (+3.4) | 46.5 (-0.7) |
>
> The VisuLogic Benchmark, due to its domain similarity, demonstrated results consistent with our experiments, whereas PuzzleVQA, due to its large domain discrepancy, conversely caused a slight degradation in performance. This performance degradation is because we performed a monolithic, single-domain fine-tuning on Qwen2.5VL, leading to a decline in the model's performance when dealing with other domains.
>
> In practice, training by blending the VisuRiddles synthetic data with the model's general SFT data is the critical solution for addressing this phenomenon, thereby allowing different domain capabilities to reinforce each other. We conducted experimental exploration using another larger-scale internal commercial business model. After blending the VisuRiddles synthetic data with the general SFT data, the model's performance not only stabilized but also saw improvement across all domains:
>
> | Train Data | InfoVQA[1] | MathVision[2] | MMMU pro[3] | Marvel (AVR)[4] |
> | :--- | :---: | :---: | :---: | :---: |
> | General SFT Data | 82.4 | 33.2 | 45.3 | 30.0 |
> | Blended Data (General + VisuRiddles) | 83.3 (+0.9) | 35.0 (+1.8) | 47.8 (+2.5) | 34.2 (+4.2) |
>
> Given that the current VisuRiddles Synthesizer primarily covers the Chinese Civil Service Examination, RAVEN, and Sudoku, we cannot directly validate it on other AVR tasks with vast sample discrepancies. Therefore, to explore PAVR's applicability in such tasks, the coverage scope of the VisuRiddles Synthesizer needs to be further expanded. Fortunately, the VisuRiddles Synthesizer, due to its modular design, is easily scalable, and we will therefore continuously optimize and improve the VisuRiddles Synthesizer in future research.
>
> [1] Mathew M, Bagal V, Tito R, et al. Infographicvqa[C]//Proceedings of the IEEE/CVF Winter Conference on Applications of Computer Vision. 2022: 1697-1706.
>
> [2] Awais M, Ahmed T, Aslam M, et al. Mathvision: An accessible intelligent agent for visually impaired people to understand mathematical equations[J]. IEEE Access, 2024.
>
> [3] Yue X, Zheng T, Ni Y, et al. Mmmu-pro: A more robust multi-discipline multimodal understanding benchmark[C]//Proceedings of the 63rd Annual Meeting of the Association for Computational Linguistics (Volume 1: Long Papers). 2025: 15134-15186.
>
> [4] Jiang Y, Zhang J, Sun K, et al. Marvel: Multidimensional abstraction and reasoning through visual evaluation and learning[J]. Advances in Neural Information Processing Systems, 2024, 37: 46567-46592.

---

> ### Author Response · Authors · 2025-11-24
> **Response to Reviewer TJrV [2 of 2]**
>
> $\textbf{Response to W3:}$
>
> We sincerely thank the reviewer for this highly insightful suggestion regarding the "rethinking" phenomenon. We agree that quantifying this behavior is essential for validating the contribution of the RL stage. To explore the value of this unique phenomenon, we utilized a commercial API (GPT-5) as a judge to determine if the SFT and RL stage outputs exhibited explicit "rethinking" behavior.
>
> We then defined the following metrics:
>
> (1) Perceptual Correction Frequency: $F_{corr} = \frac{N_{rethink}}{N_{total}} $
>
> (2) Correction Success Rate: $S_{corr} = \frac{N_{correct}}{N_{rethink}} $
>
> Where $ N_{total}$ is total number of evaluated samples, $ N_{rethink}$ is number of samples showing 'rethinking' behavior and $ N_{correct}$ is number of correctly answered samples showing 'rethinking'.
>
> The experimental results are shown in the following Table:
> | Model | $N_{rethink}$ | $\text{F}_{\text{corr}}$ |  $N_{correct}$ | $\text{S}_{\text{corr}}$ | $N_{total}$ |
> | :--- | :---: | :---: | :---: | :---: | :---: |
> | PAVR-SFT | 240 | 24% | 75 | 31.3% | 1000 |
> | PAVR | 300 | 30% | 126 | 42.0% | 1000 |
>
> The RL phase successfully enhanced the model's self-correction mechanism, increasing the frequency of "rethinking" ($24\% \rightarrow 30\%$) and, more importantly, raising the correction success rate ($\text{S}_{\text{corr}}$) by over 10 percentage points ($31.3\% \rightarrow 42.0\%$). RL not only boosts the frequency of self-checking, but also dramatically improves the effectiveness of error recovery.
>
> Furthermore, to quantify the unique robustness contribution of the RL, we performed a detailed analysis of the successfully corrected cases. Specifically, 71 unsolved error cases from PAVR-SFT were successfully corrected, precisely because the RL stage successfully learned the "rethinking" mechanism, which enabled the rectification of erroneous perception within PAVR-SFT.
>
> $\textbf{Response to Q1:}$
>
>  The perceptual descriptions are extracted directly from the synthesizer's metadata during rendering rather than predicted by the API (as detailed in Appendix B.2), ensuring 100% accuracy without perceptual noise. The API is used solely to generate CoT data based on these clean text descriptions rather than abstract graphics, eliminating visual hallucinations. We fully agree that noisy annotations can harm training.
>
> To prevent logical noise, we implement a strict Answer Validation mechanism: we compare the API-predicted answer with the Ground Truth. Only CoT samples that yield the correct answer are retained, while inconsistent ones are automatically discarded to ensure the purity of the training data.
>
> $\textbf{Response to Q2:}$
>
> We believe that both perceptual enhancements and reasoning optimization are indispensable for solving AVR tasks. However, fine-grained perception serves as the fundamental prerequisite. Taking Gemini-2.5-Pro as an example, there is no doubt that it possesses exceptional reasoning capabilities, yet it still fails to directly complete our annotation tasks due to perceptual bottlenecks.
> The ablation study in Table 4 further confirms this viewpoint, as shown below:
>
> | Model | SFT | RL | Acc |
> | :--- | :---: | :---: | :---: |
> | Baseline (Qwen2.5-VL) | × | × | 24.6 |
> | Baseline + Caption | ✓ | × | 33.3 (+8.7) |
> | Baseline + GRPO | × | ✓ | 29.4 (+4.8) |
> | Baseline + CoT (PAVR-SFT) | ✓ | × | 39.5 (+14.9) |
> | Baseline + CoT + GRPO (PAVR) | ✓ | ✓ | 46.8 (+22.2) |
>
> Applying GRPO directly to enhance reasoning without the perceptual foundation provided by SFT yields limited improvement (+4.8%), as the model struggles to generate valid outputs during training. In contrast, merely applying SFT on structured perceptual descriptions ("Baseline + Caption") yields a higher 8.7% improvement, even with the limited generalization of fixed templates.

---

### Official Review · Reviewer_emm3 · 2025-10-30

**Soundness:** 4
**Presentation:** 4
**Contribution:** 3
**Rating:** 8
**Confidence:** 4

**Summary:**

This work presents VisuRiddle, a benchmark + data synthesis framework that is designed to improve MLLMs’ capability on Abstract Visual Reasoning (e.g. RAVEN-style IQ Tests and Sodoku). The authors also provide a baseline method, PAVR, for VisuRiddle. PAVR involves a 2-stage training - first using SFT for more accurate perception, then using GRPO for more optimized reasoning. The proposed PVAR baseline model, built on top of Qwen2.5VL-7B, tops the VisuRiddle leaderboard over many larger open source MLLMs as well as common proprietary models.

**Strengths:**

I believe the authors have presented a project in high completeness. This manuscript offers clear descriptions about the motivations, the method designs, and the overall contributions.

**Weaknesses:**

My only (and perhaps relatively trivial) concern is regarding the generalizability of the new PAVR baseline method. Although PAVR is shown to offer a high bar for the self-made VisuRiddle benchmark, how does PAVR perform on existing AVR benchmarks? So far, the authors have only shown its performance on VisuLogic in Appendix D. But I believe PAVR needs to be further tested on alternative benchmarks regarding its consistency, such as LogicVista and/or VOILA for rigorous logic reasoning, and MathVerse and SeePhys for abstract perception.

**References**

- LogicVista: https://github.com/Yijia-Xiao/LogicVista

- VOILA: https://huggingface.co/datasets/nlylmz/VOILA

- MathVerse: https://github.com/ZrrSkywalker/MathVerse

- SeePhys: https://huggingface.co/datasets/SeePhys/SeePhys

**Questions:**

Since the SFT and GRPO stages to tune PVAR involve using synthesized instances created by the VisuRiddle pipeline, I would like to confirm that none of these instances used in training overlap with those in the actual VisuRiddle benchmark during evaluation.

---

> ### Author Response · Authors · 2025-11-24
> **Response to Reviewer emm3**
>
> We sincerely thank the reviewer for their careful reading and positive assessment of our work.
>
> $\textbf{Response to W1:}$
>
> Thank you for carefully reading and raising this critical question regarding the generalization of AVR tasks. However, directly validating PAVR's generalizability on other AVR benchmarks with vastly different sample domains is challenging. It is essential to clarify that our core contribution lies in verifying that perception is the primary bottleneck for MLLMs and successfully tackling the AVR problem within similar domains (i.e., similar simple rules and their combinations). Even within AVR tasks, different benchmarks exhibit significant discrepancies in visual style and inherent rules. We provide a performance comparison between the Baseline and PAVR on PuzzleVQA, which has a large domain discrepancy with data generated by the VisuRiddles Synthesizer, and VisuLogic, which has a similar domain, as shown in the table below:
>
> | Model | VisuLogic | PuzzleVQA |
> | :--- | :---: | :---: |
> | Baseline | 26.0 | 47.2 |
> | PAVR | 30.6 (+3.4) | 46.5 (-0.7) |
>
> The VisuLogic Benchmark, due to its domain similarity, demonstrated results consistent with our experiments, whereas PuzzleVQA, due to its large domain discrepancy, conversely caused a slight degradation in performance. This performance degradation is because we performed a monolithic, single-domain fine-tuning on Qwen2.5VL, leading to a decline in the model's performance when dealing with other domains.
>
> In practice, training by blending the VisuRiddles synthetic data with the model's general SFT data is the critical solution for addressing this phenomenon, thereby allowing different domain capabilities to reinforce each other. We conducted experimental exploration using another larger-scale strong internal commercial business model. After blending the VisuRiddles synthetic data with the general SFT data, the model successfully integrated stronger AVR capabilities, which resulted in performance stabilization and overall improvement across all domains:
>
> | Train Data | InfoVQA[1] | MathVision[2] | MMMU pro[3] | Marvel (AVR)[4] |
> | :--- | :---: | :---: | :---: | :---: |
> | General SFT Data | 82.4 | 33.2 | 45.3 | 30.0 |
> | Blended Data (General + VisuRiddles) | 83.3 (+0.9) | 35.0 (+1.8) | 47.8 (+2.5) | 34.2 (+4.2) |
>
> Given that the current VisuRiddles Synthesizer primarily covers the Chinese Civil Service Examination, RAVEN, and Sudoku, we cannot directly validate it on AVR tasks with vast sample discrepancies. Therefore, to explore PAVR's applicability in such tasks, the coverage scope of the VisuRiddles Synthesizer needs to be further expanded, discussed in Section 5. Fortunately, the VisuRiddles Synthesizer, due to its modular design, is easily scalable, and we will therefore continuously optimize and improve the VisuRiddles Synthesizer in future research.
>
> [1] Mathew M, Bagal V, Tito R, et al. Infographicvqa[C]//Proceedings of the IEEE/CVF Winter Conference on Applications of Computer Vision. 2022: 1697-1706.
>
> [2] Awais M, Ahmed T, Aslam M, et al. Mathvision: An accessible intelligent agent for visually impaired people to understand mathematical equations[J]. IEEE Access, 2024.
>
> [3] Yue X, Zheng T, Ni Y, et al. Mmmu-pro: A more robust multi-discipline multimodal understanding benchmark[C]//Proceedings of the 63rd Annual Meeting of the Association for Computational Linguistics (Volume 1: Long Papers). 2025: 15134-15186.
>
> [4] Jiang Y, Zhang J, Sun K, et al. Marvel: Multidimensional abstraction and reasoning through visual evaluation and learning[J]. Advances in Neural Information Processing Systems, 2024, 37: 46567-46592.
>
> $\textbf{Response to Q1:}$
>
> There is no overlap between the two sets. The basic tasks of VisuRiddles benchmark are collected from real-world exams, whereas the training instances are exclusively generated by our Synthesizer. As noted in Sec. 3.2, the synthesized data is designed to enhance perceptual capability, so its reasoning complexity is deliberately kept lower than that of the real-world benchmark.
>
> For the high-level tasks, we maintain this non-overlap rigorously:
>
> 1) RAVEN: The test set is a blend designed for rich complexity and balanced difficulty. One part is collected using open-source community generation protocols to ensure diverse, and the other part uses our self-synthesized high-difficulty instances to ensure a difficulty ceiling.
>
> 2) Sudoku: We ensured that all Sudoku instances in both the training set and the test set have unique IDs, confirming that the two sets do not overlap.

---

> > ### Comment · Reviewer_emm3 · 2025-11-25
> > **Appreciate the response.**
> >
> > I appreciate the detailed clarifications. It's reassuring to see solid improvements over multiple existing benchmarks by blending in the synthesized outputs of the VisuRiddle's data generator in post training pipelines. That being said, I believe my concern has been properly resolved.

---

### Author Response · Authors · 2025-11-25
**General response**

We sincerely thank all reviewers for their thorough reading and constructive feedback on our manuscript. We have prepared a detailed, point-by-point response addressing every weakness and question raised.

The manuscript has been revised and updated accordingly. All changes in the text of the revised manuscript are marked in blue for easy identification.

---

### Meta-Review · Area_Chair_ZBD8 · 2026-01-11

**Summary:**

The main concerns from the reviewers are following:

- Reviewer **emm3**:
  - **W1**: The generalization ability to other abstract visual reasoning (AVR) benchmarks.

- Reviewer **TJrV**:
  - **W1**: Generalization ability to real-world AVR tasks.
  - **W2**: The “rethinking” phenomenon observed in PAVR is intriguing but underexplored.
  - **W3**: More quantitative evidence to justify that PAVR’s improvements are more from perceptual enhancements than reasoning policy optimization.

- Reviewer **VP3M**:
  - **W1**: Justifying the performance bottleneck is indeed the perception descriptions still being insufficiently comprehensive other than the inherent limitations of the models' reasoning capabilities.
  - **W2**: The trained model should also be evaluated on a suite of standard MLLM benchmarks.

**Reviewer Concerns:**

- **Concerns addressed in the rebuttal:**
  - In my view, the author responses properly address all major concerns from the reviewers.

- **Concerns remained outstanding:**
  - There is no outstanding concerns.

**Reviewer Scores:**

After author responses, I think the reviewers would achieve the agreement that the paper is beyond the acceptance threshold.

---

### Decision · Program_Chairs · 2026-01-26

Accept (Poster)